

# Responses of Pine Island and Thwaites Glaciers to Melt and Sliding Parameterizations

Ian Joughin[1], Daniel Shapero[1], and Pierre Dutrieux[2]

[1]Applied Physics Laboratory, University of Washington, Seattle, 98105, USA

[2]British Antarctic Survey, High Cross, Madingley Road, Cambridge, CB3 0ET, United Kingdom

*Correspondence to*: Ian Joughin (irj@uw.edu)

**Abstract.** Pine Island and Thwaites glaciers are the two largest contributors to sea level rise from Antarctica. Here we examine the influence of basal friction and melt in determining projected losses. We examine both Weertman and Coulomb friction laws with explicit weakening as the ice thins to flotation, which many friction laws include implicitly via the effective pressure.

We find relatively small differences with the choice of friction law (Weertman or Coulomb) but find losses are highly sensitive to the rate at which the basal traction is reduced as the area above the grounding line thins. Consistent with earlier work on Pine Island Glacier, we find sea level contributions from both glaciers vary linearly with the melt volume averaged over time and space, with little influence from the spatial or temporal distribution of melt. Based on recent estimates of melt from other studies, our work simulations suggest that melt-driven combined sea-level rise contribution from both glaciers is unlikely to

exceed 10 cm by 2200. We do not include other factors, such as ice shelf breakup that might increase loss, nor factors such as increased accumulation and isostatic uplift that may mitigate loss.

## 1 Introduction

Most of Antarctica's contribution to sea level originates from West Antarctica (Otosaka et al., 2023), where ice loss occurs predominately from Pine Island and Thwaites glaciers (Rignot et al., 2019). These losses are a response to increased ocean

melting of the glaciers' buttressing ice shelves (Payne et al., 2004; Shepherd et al., 2004; Rignot and Jacobs, 2002). This enhanced melt is caused by increased transport of warm circumpolar deep water (CDW) to the glaciers' deep grounding lines (Thoma et al., 2008; Dutrieux et al., 2014; Jenkins et al., 2016), potentially in response to atmospheric forcing originating in equatorial regions (Dutrieux et al., 2014; Holland et al., 2019; Naughten et al., 2022).

Many numerical modelling studies reveal these glaciers will lose mass over the coming decades to centuries in response to continued melt forcing (Joughin et al., 2010, 2014; Seroussi et al., 2017, 2020; Favier et al., 2014). For Pine Island Glacier (PIG) at least, the amount of future ice loss appears to be a linear function of the spatio-temporally averaged sub-shelf melt rate (Joughin et al., 2021b), which is consistent with the results from a large suite of models forced with the Coupled Model



Intercomparison 6 (CMIP6) output for the Ice Sheet Model Intercomparison for CMIP6 (ISMIP6) exercise (Seroussi et al.,
2020). If further work continues to find that ice loss from well-buttressed glaciers is almost completely determined by average
melt rates, it will support a linear-response approach to projecting sea-level rise (Levermann et al., 2019).

An important control on modelled ice stream dynamics is the basal friction law, which relates basal shear stress, $\tau_b$, to the
speed, $u_b$, at which the ice slides over its bed. Early work suggested linear viscous behaviour for soft (weak till) beds
(Blankenship et al., 1987) and Weertman sliding (power law with an exponent of ~3) over a hard bed (Weertman, 1957).  Later
work showed that weak till beds are far better approximated as Coulomb-plastic behaviour (Kamb, 1991; Zoet and Iverson,
2020). Moreover, when cavitation effects are incorporated in sliding models, Coulomb-like conditions should occur for fast
basal sliding over hard beds (Gagliardini et al., 2007; Schoof, 2005). Thus, both hard and soft beds may be well represented
by a Coulomb model, at least at fast sliding speeds (Minchew and Joughin, 2020). Historically, models used to project ice
sheet loss over the next few centuries use models ranging from linear-viscous to Coulomb plastic, or some hybrid combination
(Asay-Davis et al., 2016). While the sliding coefficient for such parameterizations typically can be solved for with inverse
methods (MacAyeal, 1993), other factors such as the exponent and the treatment of effective pressure are less well constrained.

Here we examine the sensitivity of the responses of Pine Island and Thwaites glaciers to a) various parameterizations of the
friction law, and b) the mean aggregate basal melt for their respective ice shelves.  In particular, we focus on a regularized
Coulomb friction (RCF) law that prior work indicates best replicates recent observations of Pine Island Glacier (PIG) (Joughin
et al., 2019).

## 2. Basal Friction Overview

One of the more widely used forms of the friction law is the power-law relation

$$\boldsymbol{\tau_b} = \beta_m^2 |\boldsymbol{u_b}|^{\frac{1}{m}} \frac{\boldsymbol{u_b}}{|\boldsymbol{u_b}|}, \qquad (1)$$

which is often used with a value of $m=3$ to produce Weertman sliding (Weertman, 1957). As $m$ becomes very large this relation
tends toward Coulomb basal friction, which can be expressed as

$$\boldsymbol{\tau_b} = \beta^2 \frac{\boldsymbol{u_b}}{|\boldsymbol{u_b}|}. \qquad (2)$$

Following the convention of MacAyeal (1993), the friction coefficient in these equations is expressed as $\beta^2$ to ensure a positive
value is determined when using inverse methods.






Work by Schoof (2005) and Gagliardini et al. (2007) indicates that while Weertman conditions can occur at slow speeds, at high speeds water-filled cavities form in the lee of basal bumps, causing more Coulomb-like or even velocity-weakening behaviour. Based on this work, large-scale ice-sheet-modelling studies often use a basal friction law with the form (Asay-Davis et al., 2016)

$$\tau_b = \frac{\beta^2 |u_b|^{\frac{1}{m}} \alpha^2 N}{(\beta^{2m} |u_b| + (\alpha^2 N)^m)^{\frac{1}{m}}} \frac{u_b}{|u_b|}, \qquad (3)$$


where $\alpha^2$ is the Coulomb friction coefficient (typically 0.5), and $N$ is the effective pressure, which is the difference between the overburden and basal-water pressures. Tsai et al (2017) developed an alternative expression to combine Weertman and Coulomb behaviour as

$$\tau_b = \min(\alpha^2 N, \ \beta^2 |u_b|^{\frac{1}{m}}) \frac{u_b}{|u_b|}. \qquad (4)$$


Equations (3) and (4) both depend on the effective pressure, $N$. An often-used convention is to assume a perfect hydrological connection to the grounding line so that the basal water pressure equals the ocean pressure (e.g., Asay-Davis et al., 2016). In this case, the effective pressure is given by


$$N = \rho_i g (h - h_f), \qquad (5)$$

where $\rho_i$ is the density of ice, $g$ is the acceleration due to gravity, $h$ is the ice thickness, and $h_f$ is the flotation height (the elevation at which ice begins to float).


Figure 1 illustrates the sensitivity of these friction laws to speed for parameters meant to represent the near-grounding line region, central trunk, and outlying tributaries of PIG. In these examples, Weertman conditions are found everywhere except where $h_f$ equals 45 meters. The transition from Coulomb to Weertman conditions in Equation (4) occurs at $h - h_f = \frac{\tau_b}{\alpha^2 \rho_i g}$, with Equation (3) producing a less abrupt transition at a similar value. Thus, if we assume 300 kPa as the maximum expected value for $\tau_b$, then the transition to Weertman sliding takes place at locations where the elevation is less than 67 m above flotation for $\alpha^2$=0.5. To illustrate the extent of the region where Coulomb conditions occur with these models, Figure 2 shows contours of $h - h_f$ plotted over values of $\tau_b$ inferred as described below. These contours indicate that Coulomb conditions only occur within about 10 km of the grounding line, which is consistent with the distances over which the assumption of perfect hydrological connectivity is likely valid (Asay-Davis et al., 2016).




Numerous boreholes indicate water pressures close to flotation, and thus, low (<400 kPa) effective pressure (Luthi et al., 2002; Kamb, 2001) well away from the grounding line. The widespread presence of active subglacial lakes also suggests that low effective pressures are prevalent (Gray et al., 2005; Smith et al., 2009; Fricker et al., 2007; Bell, 2008). Thus, if we were able to accurately model effective pressure away from the grounding lines, the results likely would indicate that extensive areas of

low effective pressure exist well inland of the grounding line. If this is the case, then Equations (3) and (4) indicate Coulomb conditions occur over a much broader area than is indicated by assuming basal water pressure equals ocean pressure.

Initially based on laboratory measurements by Budd et al. (1979) and later modified by Fowler (1987), a modification of Weertman sliding that adds an explicit effective pressure dependence is given by


$$\boldsymbol{\tau_b} = \beta_m^2 \mathrm{N}^{\frac{q}{m}} |\boldsymbol{u_b}|^{\frac{1}{m}} \frac{\boldsymbol{u_b}}{|\boldsymbol{u_b}|}. \qquad (6)$$

This equation has been termed Budd friction and it is often applied with the effective pressure as given by Equation (5), even at large distances from the grounding line where the assumption is likely not valid (Yu et al., 2018).

We know of no basal hydrological models for $N$ that have been demonstrated to have sufficient accuracy with which to determine basal shear stress, leading to the often-used assumption given in Equation (5). An alternate approach is to assume that effective pressures are low enough in regions of fast flow to produce Coulomb conditions. In this case, the observed speeds are assumed to determine the extent and type of basal friction with the transition from Weertman to Coulomb behaviour occurring at some critical speed, $u_o$. By subsuming the effective pressure into the sliding coefficient ($\beta^2$), the form of the

equation given by Schoof (2005) can be rewritten (Joughin et al., 2019) as

$$\boldsymbol{\tau_b} = \beta^2 \left( \frac{|u|}{|u| + u_0} \right)^{\frac{1}{m}} \frac{\boldsymbol{u_b}}{|\boldsymbol{u_b}|}, \qquad (7)$$

which we refer to as regularized Coulomb friction (RCF). In this case, the influence of $N$ is determined implicitly in the solution for $\beta^2$. Although this expression was derived for sliding-induced cavitation on hard beds (Schoof, 2005; Gagliardini et al.,

2007), laboratory studies show this form also applies to soft, deforming beds (Zoet and Iverson, 2020), albeit with a potentially different exponent. Thus, it may be reasonable to model friction with a single friction law (Minchew and Joughin, 2020). When modified as described below and included in a model forced with observed elevation change, this friction model with m=3 most accurately reproduced the observed speedup of PIG over nearly two decades (Joughin et al., 2019). As indicated in Figure 1, this equation produces Coulomb-like friction in regions of fast flow ($|u| > u_o$) and Weertman-like conditions in areas

of slower flow ($(|u| < u_o$; Figure 1c). Another study indicates PIG conditions are reproduced better with values of $m$ in the range of 10–20, which produces a sensitivity of $\tau_b$ to speed that more closely resembles that of Equation (7) than that of Weertman sliding (Gillet-Chaulet et al., 2016).





In addition to determining the transition from Weertman to Coulomb conditions, the sensitivity to $h - h_f$ for the assumed form

of $N$ used in Equations (3) and (4) causes the bed to weaken as the ice approaches flotation. This is a desirable effect since it

is unlikely that the basal traction goes from full strength to nothing precisely as the ice base goes from grounded to floating.

To illustrate this point, Figure 2c shows the cumulative distribution functions (CDFs) for values of $\tau_b$ inferred as described

below for height-above-flotation-determined bands near the grounding line. The results show that near the grounding line ($h -$

$h_f < 41$ m) low $\tau_b$ values (median 66 kPa) are far more likely than ~5–15 km farther inland ($41 < h - h_f < 176$ m) where

higher values of $\tau_b$ (medians of 97 and 125 kPa) are more prevalent. This characteristic is consistent with observations of a

break in slope near the grounding line, indicating reduced driving and basal-shear stresses (Fricker and Padman, 2006). These

observations suggest that as the grounding line (zone) recedes inland to areas that presently are strong, some reduction in basal

traction occurs.

Figure 3 shows the weakening as the surface elevation approaches flotation for Equations (3) and (4). The difference in these

two formulations is that Equation (3) combines Weertman and Coulomb basal resistance in parallel (Gudmundsson et al.,

2023), which provides a smoother transition between the two friction types than does the more abrupt transition when using

Equation (4).

The Weertman and RCF models as parameterized above have no such weakening. To include this effect, Joughin et al. (2010)

included a linear weakening of the bed that initiates once the height above flotation falls below some threshold, $h_T$, which can

be expressed as

$$\lambda(\text{h}) = \begin{cases} 1 & (h - h_f) > h_T \\ \frac{(h - h_f)}{min(h_T, \ h_0 - h_f)} & (h - h_f) \le h_T \end{cases}, \quad (8)$$


where $h_o$ is the elevation at the start of the simulation. When used to scale (e.g., $\boldsymbol{\tau_b}(h) = \lambda(h)\boldsymbol{\tau_b}$) the results in Equations (1)

or (7), this function produces linear weakening similar to Equation (4) as shown in Figure 3. There is a critical difference,

however, in that $h_T$ in Equation (8) is fixed for all values of $\tau_b$, whereas the elevation-dependent weakening in Equations (3)

and (4) occurs with a spatially varying threshold determined by $\tau_b$. The former assumes some critical threshold for effective

pressure, which applies to a range of bed conditions. That latter assumes an effective pressure limit that depends on how much

shear stress the bed can support. Each represents imperfect assumptions, and it's not clear which model is preferable in the

absence of a better solution.





Earlier work suggested that a value of $h_T$=41–46 m best reproduces PIG's response over the last two decades (Joughin et al.,
2019, 2021a) for the RCF model (Weertman sliding produced best results with $h_T = 123$). Figure 2, however, suggests the
weakening may occur over a broader zone (see Figure 2c). As a result, a major goal of the work presented here is to investigate
the sensitivity of losses projected over centuries to this parameter. Here we perform experiments using Equation (8) because
it allows us to vary the amount of weakening so that we can study the resulting impact on ice loss. It also lets us separate the
weakening behaviour from the friction model. Thus, many of the experiments described below are aimed at understanding the
sensitivity of ice loss to the choice of $h_T$.

## 2 Methods

Our numerical experiments are all conducted using a finite element modelling package called *icepack* (Shapero et al., 2021).
The remainder of this section describes the setup and initialization used to conduct the experiments described below.

### 2.1 Model

Our results are based on simulations using a basin-scale model of a coupled ice-sheet/shelf system that was developed for
earlier studies of PIG (Joughin et al., 2021a, b). The ice-sheet modelling package, *icepack* (Shapero et al., 2021), used to
construct this model is built around the finite-element analysis library, Firedrake (Rathgeber et al., 2016), which includes an
embedded symbolic language for specifying the differential equations to be solved. Both *icepack* and Firedrake are fully open
source and are available through GitHub as is the basin-scale model used for the work described here.


The model solves the shallow-shelf equations (MacAyeal, 1989) on an unstructured finite-element mesh with triangular
elements. The mesh spacing is variable, with resolutions of a few hundred meters near the grounding lines and several
kilometres in the deep interior (Figure S1). The model does not account for glacial isostatic adjustment since this effect should
be small at the 200-year time scales examined here (Larour et al., 2019). A typical 200-year simulation for the combined
basins of Pine Island and Thwaites glaciers takes a few days on a single CPU core with first-order elements and a time step of
0.01 years. Since everything is single-threaded, we can run ensembles in parallel on a multi-core machine over the same period.
(Larour et al., 2019)

Rather than using Equation (6) to implement RCF, within *icepack*, it is numerically easier to solve for a different expression
given by (Shapero et al., 2021)

$$\tau_b = \frac{\beta^2 |u_b|^{\frac{1}{m}}}{\left(u_o^{\frac{1}{m}+1} + |u_b|^{\frac{1}{m}+1}\right)^{\frac{1}{m+1}}} \frac{u_b}{|u_b|}. \qquad (9)$$





We refer to the icepack version of regularized Coulomb friction as RCFi. Although Equations (7) and (9) appear substantially different in form, by adjusting the values of $u_o$, they can produce nearly equivalent responses, as demonstrated in Figure 1 (compare RCF and RCFi with values of $u_o$ equal to 200 and 300 m yr$^{-1}$ in Figure 1, respectively).


We initialize the model by inverting for the basal friction law parameters using standard methods implemented in *icepack*, with the amount of regularization determined through L-curve analysis (Hansen and OLeary, 1993). The inversion procedure also solves for the Glen's flow law parameter, *A*, on the floating ice. For the grounded ice the model determines *A* based on its temperature dependence (Cuffey and Paterson, 2010) using an earlier simulation for internal temperature (Joughin et al., 2009).


For most of the experiments, we use a randomly generated ensemble of 30 melt distributions (Joughin et al., 2021b). The distributions are selected such that approximately half produce peak melt at the grounding line while the other half produce peak melt higher in the water column. At each time step, each melt distribution is re-normalized to produce a specified level of melt. By contrast, many studies use a melt function parameterized by depth (e.g., Joughin et al., 2014; Gudmundsson et al., 2023; Jourdain et al., 2020; Yu et al., 2018; Barnes and Gudmundsson, 2022). In addition to our melt ensembles, we also conducted simulations with depth-parameterized melt rates used in other recent studies (Gudmundsson et al., 2023; Yu et al., 2018; Barnes and Gudmundsson, 2022).

### 2.2 Initialization Data Sets

Our study area is the combined basins of PIG and Thwaites glaciers. Note that we treat Haynes Glacier as a branch of Thwaites Glacier so that all references hereafter to Thwaites apply to both glaciers. For the surface and bed elevations and thickness, we used the BedMachine Version 3 data set (Morlighem et al., 2020). We modified the bed elevations slightly to make them consistent with other data. First, we reduced bed elevations along some areas of the ice front to ensure they were floating, consistent with the assumed boundary condition. Second, we raised elevations for a few pockets in the interior to ensure that they were grounded. Finally, we reduced elevations downstream of the grounding line position of PIG that we inferred from 2014 TerraSAR-X speckled-tracked offsets (Joughin et al., 2016), which agrees well with the MEaSUREs Version 2 grounding line (Rignot et al., 2014).

To invert for the friction coefficient, we used a velocity map assembled from three sources. First, we produced a map by processing all the Sentinel 1A/B data for the region collected from January 2019 to December 2020, which covered most of the model domain north of 78.7°S. Next, we filled gaps using data from the MEaSUREs Phase-Based Velocity (Version 1) map (Mouginot et al., 2019). Finally, for some of the slower, southernmost regions where there were gaps or excessive noise in the MEaSUREs data, we used balance velocity data computed with a well-established algorithm (LeBrocq et al., 2006) constrained by RACMO 2.3 (Wessem et al., 2014) surface mass balance (SMB) data averaged from 1979-2022. Because most of the fast-moving areas are covered by the Sentinel-1 data, the final map is representative of the mean flow for 2019-2020.






For the surface mass balance (SMB), consistent with earlier studies (Joughin et al., 2021b, 2014), we used a map of SMB derived from airborne radar and ice cores (Medley et al., 2014). Using this SMB and initializing the model with the observed velocities, the combined system initially loses ice at a rate of 0.33 mm yr$^{-1}$ sea level equivalent (sle), with PIG and Thwaites losing 0.17 and 0.16 mm yr$^{-1}$ sle, respectively. Note all values presented here are computed on a polar stereographic grid,

which introduces area distortion. As a result, our estimated losses are biased low by ~2.5%.

**3.0 Results**

Earlier work simulated the response of PIG to melt rate using $h_T$=41 m (Joughin et al., 2021b), which is our preferred value since it provides the best match to PIG's recent behaviour using RCFi (Joughin et al., 2019). The PIG record is short (<2 decades), however, relative to the periods over which sea-level projections are required (centuries).  To examine the sensitivity

of simulated losses to the choice of $h_T$ and melt rate, we conducted further simulations with an expanded domain that also includes Thwaites Glacier.

Figure 4 shows the volume above flotation (VAF) losses for 200-year simulations with RCFi and our preferred value of $h_T$ (41 m) and for melt rates of 57, 75, 100, and 125 Gt yr$^{-1}$ applied to each ice shelf (i.e., total melt for the domain is twice these

values).  For PIG the results agree well with those from earlier work (Joughin et al., 2021b), with small differences due to differences in the initial conditions and interactions with the adjacent Thwaites basin. Thwaites and PIG produce similar losses throughout these simulations when forced with the same melt levels. Also shown are the combined VAF losses that would occur if the current rates are linearly extrapolated. The combined 200-year VAF losses are 73% and 85% of extrapolated current rates for melt rates of 57 and 75 Gt yr$^{-1}$, respectively. With the higher melt rates, losses exceed the present rates by 7–

28%. As with the rest of the simulations described here, the results represent an average of 30 randomly selected melt profiles, each normalized to produce the prescribed amount of melt as described above.

Figure 5 shows combined losses for both glaciers for the four values of $h_T$ that correspond to the height-above-flotation contours shown in Figure 2. The simulations were conducted using both RCFi and Weertman sliding. For both types of friction,

the VAF loss is strongly sensitive to the choice of $h_T$. For the least melt (57 Gt yr$^{-1}$), the largest value ($h_T$=172 m) produces ~40% more loss than the smallest value ($h_T$=1 m). At the highest melt rate (125 Gt yr$^{-1}$), the corresponding difference is more than a factor of 2. The behaviour is similar for the Weertman cases, except that the sensitivity to $h_T$ is substantially lower at the lower end of the melt range. At the two lowest melt rates, nearly all the simulations produce less loss than extrapolation of the current rate. By contrast, the simulated losses exceed the extrapolated current rate with the two highest melt rates for all

but some of the cases with $h_T$=1 m.



Figure 6 shows the annual loss rates for each glacier from the RCFi simulations. These results are averages of 30 simulations with differing melt distributions, so they tend to smooth out the variability of individual ensemble members as the grounding line retreats off basal highs (Joughin et al., 2021b). After a brief initial transient as the system equilibrates to the imposed melt, the VAF losses occur at relatively steady rates throughout most of the simulations. For Thwaites with the larger $h_T$ values, however, the annual rates of loss tend to increase substantially (~2-3x) throughout the simulation. At the most extreme ($h_T$=172 m and 125 Gt yr$^{-1}$ melt), the end-of-simulation loss rate for Thwaites is more than 5 times the present rate and more than twice the rate for PIG. As a result, much of the sensitivity to $h_T$ for the combined losses shown in Figure 5 is attributable to Thwaites Glacier.

To examine the sensitivity to melt, Figure 7 shows the 200-year losses as a function of the melt rate for the combined and individual glacier basins. To illustrate the sensitivity to the different spatial distributions of melt, this figure also shows the individual melt-distribution ensemble members for the RCFi simulations. Linear fits to raw ensemble data (120 points for each fit) for both friction models are also shown. For the combined basin, the regressions show that the melt rate accounts for most of the variance (88–94%), with the remaining variance due to the spatial distribution. The corresponding ranges are 81–97% and 62–92% for PIG and Thwaites glaciers, respectively. The sensitivity to melt increases with $h_T$ as indicated by the slopes for the combined RCFi responses, which vary from -0.21 to -0.61 mm Gt$^{-1}$ yr sle over the range $h_T$ values. The corresponding range of sensitivity for PIG is -0.24 to -0.51 mm Gt$^{-1}$ yr sle for PIG and -0.18 to -0.71 mm Gt$^{-1}$ yr sle for Thwaites. The results are similar for the Weertman cases, except that the sensitivity to $h_T$ is somewhat greater for PIG (-0.21 to -0.7 mm Gt$^{-1}$ yr sle). This increase in slope is largely due to the substantially lower losses for Weertman sliding at the lower end of the melt range.

To demonstrate their spatial distribution, Figure 8 shows the VAF loss averaged over all 30 ensemble members at each level of melt for both the RCFi and Weertman simulations with $h_T$=41 m. All the simulations have some thickening in the upper basin, which is likely due to the poorer quality of the velocity used to initialize the model there (i.e., speeds that are too slow). At the lower elevations, there is strong thinning of up to a few hundred m/yr that increases with melt level. At the higher melt values, the results from RCFi and Weertman are similar. At the lower melt levels, Weertman cases show some slight thickening near the grounding line and less overall thinning, consistent with the results shown in Figure 7. Figure 9 shows the extent of grounding line retreat for the different melt levels and friction laws. These results show grounding line advance for the low melt Weertman cases.

**4.0 Discussion**

Our simulations of Pine Island and Thwaites glaciers reveal several important aspects of how projected contributions to sea level are influenced by the friction model, loss of traction above the grounding line as ice thins ($h_T$), and the melt rate.



## 4.1 Sensitivity to Friction Law

Overall, our results indicate the choice of friction law yields relatively minor differences to the projected VAF losses (Figure
7), except for the PIG cases with low melt. These differences are consistent with the PIG re-grounding seen in the low melt
simulations with Weertman sliding (Figure 9e&f).  In experiments to reproduce the observed velocity constrained by measured
elevation changes from 2002 to 2017, Weertman sliding tended to yield velocities that were too fast in the vicinity of the
grounding line (Joughin et al., 2019). Over time, this additional outflow should thicken the ice shelf and cause grounding-line
advance if the melt is insufficient to handle the excess discharge, which may explain what happens for our low-melt Weertman
simulations (Figure 9e&f). On Thwaites Glacier, all the simulations result in retreat and the RCFi and Weertman simulations
produce roughly similar results in terms of grounding-line retreat (Figure 9) and VAF loss (Figures 7–8). Overall, the
grounding line retreat is more variable for Thwaites (blue-green areas in Figure 9). This variability likely occurs because
approximately half the ensemble members tend to produce shallower melt distributions that shift a larger portion of the melt
to the eastern shelf, which should enhance retreat along the eastern portion of the grounding line. Thus, while the choice of the
form of the frictional law makes a relatively small difference (< 20%) in general, the friction models cannot be treated
interchangeably since in some circumstances the differences can be large (>50%) and unpredictable, as in the low-melt PIG
simulations. Since RCFi is better able to reproduce recent behaviour on PIG and because Weertman friction can cause
grounding-line advance (Figure 9) that is inconsistent with current observations, RCFi seems preferred for simulations where
only one type of friction is used.


Several studies have compared Weertman sliding with the friction laws expressed in Equations (3) and (4) (Gudmundsson et
al., 2023; Barnes and Gudmundsson, 2022; Nias et al., 2016). Comparisons of our work with those studies are hindered by the
fact that while the friction laws represented by these equations are often referred to as regularized Coulomb friction, they
produce Coulomb friction for only a small fraction of the bed (<1% of the domain) near the grounding line ($h - h_f < {\sim}86$ m
in Figure 2) so that the vast majority of the basin is subject to Weertman sliding. By contrast, RCFi applies Coulomb conditions
to the full extent of the fast-moving regions (~11% of the domain). A further complicating factor is the extent to which
Equations (3) and (4) differ from Weertman due to Coulomb behaviour (i.e., the dependence of $\tau_b$ on speed) versus their
dependence on effective pressure (i.e., reduction in traction as flotation is approached). Moreover, such a reduction is not
limited to Coulomb friction and such a dependence can be included in a Weertman model as is the case for Equation (6).


An advantage of our approach is that we can evaluate how the friction law and the weakening above the grounding line
individually affect ice loss. As the latter appears to play a larger role, we defer comparison to the discussion below where we
examine the sensitivity of our results to $h_T$.



## 4.2 Sensitivity to Weakening as Ice Approaches Flotation

Figure 2c indicates that the area near the grounding line is substantially weaker than the area immediately above it, which suggests the need for a mechanism to reduce basal traction as the ice column evolves toward flotation. This reduction can be accomplished either explicitly through Equation (8) or implicitly through the dependence on effective pressure in Equations (3) and (4). All of our simulations use the explicit approach.

The results shown in Figures 5–7 clearly indicate a strong sensitivity to the rate at which traction above the grounding line is reduced in our simulations. Larger values of $h_T$ tend to produce more loss at the same melt level due to the additional loss of basal traction as the ice thins. An exception occurs for PIG where, at low melt values, the losses are nearly the same across the full range of $h_T$. An earlier analysis of PIG indicated that while loss of traction acts to speed the glacier up, much of this effect is counterbalanced by the evolution of the surface, which reduces the driving stress near the grounding line (Joughin et al.,

2019). In most cases the loss of basal traction appears to prevail, leading to an overall speedup. At low melt rates on PIG, however, these two competing effects appear to roughly balance each other over the full range of $h_T$. Thwaites losses are far more sensitive to $h_T$, with the min and max values producing differences of ~60% at the low end of the melt range up to a factor of 2.8 at the upper end of the range. This enhanced sensitivity is likely due to its weak shelf, which is less able to produce additional buttressing as it speeds up to help compensate for the greater loss of basal traction due to high melt with a large

value of $h_T$.

As Figure 3 indicates, the loss of traction near the grounding line as the ice approaches flotation is similar in form for Equations (3), (4), and (8). The key difference is that for our simulations, the linear reduction in traction is determined by a spatially invariant threshold ($h_T$). The way Equations (3) and (4) are formulated means that they effectively have similar thresholds,

except that they vary spatially based on the basal shear stress at each point. These differences tend to make direct comparisons difficult. One way to obtain a rough equivalency is to determine the value of $h_T$ that yields equivalent area-integrated traction subject to reduction via the effective pressure dependency in Equation (3) for a given value of $\alpha^2$.

Barnes and Gudmundsson (2022) conducted simulations using $\alpha^2$ values of 0.25, 0.5, and 0.75 in Equation (3), which roughly

translates to $h_T$ values of 57, 27, and 17 m, respectively, using the method just described. These values bracket our preferred value of 41 m, which indicates this choice of $h_T$ is consistent with the range of values they used as well as values used for other studies. They conducted several 120-year simulations for a domain that also included Smith, Pope, and Kohler glaciers. When they used the friction model described by Equation (3), the results were 20 and 38% greater relative to Weertman sliding ($\alpha^2 \rightarrow \infty$) for $\alpha^2$ values of 0.5 ($h_T \sim 27$ m) and 0.25 ($h_T \sim 57$ m), respectively. Over the same period using Weertman sliding

scaled by $\lambda(h_T)$ with $h_T$=41 m, we obtained results that were 23% greater than with $h_T$=1 m (Weertman with effectively no weakening). Given the differences in the models and domains, our results are in good agreement with theirs, suggesting that




most of the additional losses in the Barnes and Gudmundsson (2022) results are due to the reduction in basal traction as the effective pressure declines rather than the transition to Coulomb conditions in the region near the grounding line.

The fact that our empirically-derived value of $h_T$ agrees well with roughly equivalent values determined from consideration of effective pressure suggests that both types of models tend to reduce basal traction at rates that are approximately the right magnitude. While we cannot completely discount the results from the larger values of $h_T$ used in our simulations, they likely produce losses that are larger than can be expected.

Consistent with the other work cited above, the results presented here suggest that models should include some type of reduction in basal traction as flotation is approached, irrespective of the actual friction type (e.g., Weertman or Coulomb). Less clear is how such weakening should be applied. While empirical in nature, Equation (8) has demonstrated a reasonable ability to reproduce observed behaviour (Joughin et al., 2019). There is no reason, however, that $\alpha^2$ in Equations (3) and (4) cannot be selected through a procedure like that used to derive our preferred value of $h_T$. On the other hand, Equation (8) can easily
be modified to have a spatial variable $h_T$ that depends on effective pressure in a similar manner to Equations (3) and (4), which would allow the traction reduction to be decoupled from the form of the basal friction law. The best combination of these concepts is a subject for future research.

As the ice evolves in areas away from the grounding line, the driving stress increases in some areas and decreases in others. It
is unclear the extent to which the friction law parameters are static, as is often assumed, or whether they should instead co-evolve with the driving stress or other changes in the ice-sheet geometry that influence effective pressure. Lacking a good model to vary the friction as the surface evolves, except near the grounding line, many models, including ours, only allow basal traction in the interior to respond to variations in speed (e.g., Seroussi et al., 2020). Budd friction with effective pressure determined by Equation (5) is an exception in that the basal traction is reduced over the entire model domain in direct response
to thinning. When using this type of friction, projected losses can more than double (Yu et al., 2018; Barnes and Gudmundsson, 2022). With q=3 and m=3 in Equation (7), the result is equivalent to Weertman friction with unbounded $h_T$ (i.e., basal traction declines linearly with reductions in $h - h_f$). The assumption, however, that a hydrologic connection to the ocean exists over the full domain such that the water pressure is equal to ocean pressure is not well supported by borehole observations of water pressure (Luthi et al., 2002; Kamb, 2001) and the widespread presence of subglacial lakes (Gray et al., 2005; Smith et al.,
2009; Fricker et al., 2007; Bell, 2008). We suggest that any law that relies solely on the local height above flotation to govern changes in effective pressure, and thus, basal friction over the entire domain is likely oversimplified and incorrect. Other factors, such as the surface slope, should influence the basal hydrological system that determines effective pressure. As a result, care should be taken in interpreting results that employ Budd friction (Equation 7) in the absence of a more accurate way of determining the effective pressure.






### 4.3 Sensitivity to Melt

Our results indicate that the combined and individual losses for PIG and Thwaites Glacier increase linearly with melt (Figure 7), consistent with similar work for just PIG (Joughin et al., 2021b). For PIG, the linear fits to the 120 ensemble members for each friction model (30 melt distributions for each of 4 melt levels) indicate that the spatiotemporal averaged melt has a far

greater effect on ice loss ($r^2$=0.86–0.97) than does either the spatial or temporal variation of the melt, consistent with an earlier study that simulated just the PIG basin (Joughin et al., 2021b). The PIG ice shelf provides substantial back stress (Gudmundsson et al., 2023). As a result, increasing the discharge to the shelf provides more backstress (faster flow and thicker ice), which acts as a negative feedback on the speed. Greater melting weakens this response, allowing greater discharge. For Thwaites Glacier, the sensitivity to melt is weaker for lower values of $h_T$ with less of the variance being explained by the trend

($r^2$=0.6–0.69). Thwaites Glacier is not nearly as well buttressed by its ice shelf as PIG (Gudmundsson et al., 2023), so the flow is less sensitive to melt-induced thinning of the shelf at low values of $h_T$, consistent with an earlier sensitivity study (Nias et al., 2016). Moreover, Thwaites has a broad weak shelf with a shallow draft and a narrow, deep pockets that provide some buttressing (Gudmundsson et al., 2023). As mentioned above some of the shallower melt distributions will concentrate more of the melt on the eastern shelf to yield more variable results.


We used a fixed melt level throughout each simulation. In other work on PIG, however, similar simulations with periodic melt forcing (periods of decades to centuries), linear melt trends, and steady melt all produced virtually the same losses in cases where the long-term average melt was the same (Joughin et al., 2021b). This finding might appear to contradict other work suggesting melt rate variability about a constant long-term average rate can affect overall VAF loss (Robel et al., 2019;

Hoffman et al., 2019). These studies, however, examine fluctuations in melt rate rather the volume, which is a function of both the melt rate and the shelf geometry. For example, a constant melt rate will lead to an increasing melt volume as the shelf area expands with ungrounding. Thus, we speculate that if these earlier results were recast as functions of melt volume, then they would exhibit a similar linearity with melt volume as do our results. While we are not well positioned to repeat these prior experiments, we can examine whether such linearity holds for melt volume that freely evolves when driven by fixed depth-

dependent melt-rate parameterizations. To do so, we conducted additional experiments using the depth-parameterized functions used in other studies (Barnes and Gudmundsson, 2022; Gudmundsson et al., 2023; Yu et al., 2018), which allow the melt to vary freely as the shelf evolves. These simplified melt functions (Figure S2) are meant to crudely emulate a plume originating in a warm bottom layer (<500–1000 m) with high melt rates (40-160m yr[-1]) that rises through a linear melt gradient approximating the thermocline at middle-depths, above which the plume loses all ability to melt ice.


Figure 10 shows the results using these 10 depth-parameterized melt rates, which produce average melt per glacier ranging from 36 to 389 Gt yr[-1], with melt on Thwaites of up to 549 Gt yr[-1]. Linear regressions to the results from these simulations produce $r^2$ values of 0.97 or greater, except for one case on Thwaites ($r^2$=0.9 for $h_T$=1). The independently determined



regressions from our constant-melt ensembles all fit nearly as well ($r^2_{const.melt}$ = 0.77–0.98). These results indicate that the linear increases with steady melt evident in Figure 7 also apply when the melt freely evolves with the shelf geometry. We note that the regressions to the depth-parameterized melt rates typically yield higher $r^2$ values (Figure 10) than do the regressions to the constant-melt ensemble data (Figure 7), which may be a statistical quirk. The better fits for the depth-parameterized functions, however, may reflect the fact that our melt ensembles employ a wider range of depth variation. For example, all the depth-parametrized melt functions produce maximum melt in the bottom part of the water column near the grounding line. By contrast, half of the ensemble distributions produce maximum melt higher in the water column as some models suggest should be the case (Favier et al., 2019)

Both our ensembles and the depth-parameterized melt simulations reveal that ice losses increase linearly with melt. Although we used a single model, the ISMIP6 suite of models yields similar results (Seroussi et al., 2020) with a linear regression of sea level rise on melt yielding $r^2$=0.93 for the Amundsen Sea Embayment (Joughin et al., 2021b). Based on a similar assumption of linearity, Leverman *et al.* (2019) characterized sea-level uncertainty from Antarctica by generating large ensemble estimates based on a more limited number of runs from ISMIP6. The linear response to melt shown in Figures 7 and 10 supports the use of their approach, which may help limit the computational burden for large ensemble projections.

Given that shelf-wide total melt is a robust predictor of sea level rise contributions, future studies should include total melt values in addition to other descriptors of melt (e.g., average melt rates or melt parameterizations) to facilitate comparison with other studies as discussed above. For example, plotting results from multiple studies as shown in Figure 9 would help differentiate the cases where different models produce results consistent with the level of melt forcing (e.g., the results lie along a line) from those where the differences are due to some other aspect of the model (e.g., results that fall well off a line). For example, the fact that melt is the main predictor of loss in the Amundsen Sea Embayment for the suite of ISMIP6 models (Seroussi et al., 2020), suggests the differences between models may largely be due to how they treat melt, as opposed to differences in their treatment of ice dynamics.

Our use of prescribed, rather than freely evolving, melt rates does not necessarily emulate natural processes. It does, however, provide a controlled way to evaluate the response of a coupled ice-sheet/ice-shelf system to melt forcing. The resulting regressions reliably predict the VAF loss in cases where the total melt can be determined (e.g., dashed curves in Figure 10). While we used a constant melt forcing, a melt history supplied by any methods that can estimate total melt for a given cavity geometry (e.g., offline ocean model) could be used. Removing the details of the spatial distribution of melt may allow the use of simpler, more loosely coupled models that only need to determine the total melt at infrequent intervals so long as they track the long-term melt trend.



## 4.4 PIG and Thwaites Outlook

Our simulations are not projections since they are not tied to climate forcings. Nor do they include factors not related to ocean melt such as increased accumulation as atmospheric temperatures rise (Donat-Magnin et al., 2021), loss of ice shelf area, or glacial isostatic adjustment (Larour et al., 2019). Nonetheless, they do provide some sense of how much these glaciers will
contribute to sea level rise over the next two centuries in response to basal melt.

Prior estimates for PIG melt range from 76 to 101 Gt/yr (Rignot et al., 2013; Depoorter et al., 2013; Shean et al., 2019; Adusumilli et al., 2020) but these estimates cover an area substantially larger than our domain. A recent melt estimate from remote sensing that covers an area similar to our model domain is 67 Gt $yr^{-1}$ with substantial interannual variability (Joughin
et al., 2021b). This value lies between our 57 and 75 Gt $yr^{-1}$ simulations both of which produce future losses less than the present rate. The current rates, however, include speedup due to recent ice-shelf loss, which is expected to decline as the system adjusts to the new geometry. Our 100 and 125 Gt $yr^{-1}$ simulations produce long-term average losses greater than present for PIG.

Recent simulations of water temperatures with regional-scale ocean models forced with climate model output indicate that melt rates on PIG will increase by ~5–8 m/yr, which is equivalent to 21 Gt $yr^{-1}$ by 2100 for the current ice-shelf geometry. If there is a similar increase for the next century, then our 125 Gt/yr estimate would still exceed the two-century average. This analysis, however, does not allow for increases in ice-shelf area (Figure 9), which also influence the melt rate. Neglecting the expansion of the ice shelf may not have a big impact on results for PIG. A coupled ice-ocean model produces a relatively
steady melt rate of ~150 Gt/yr for warm conditions (base of thermocline at 600 m) through a 120-year simulation in which the glacier has a total VAF loss of 50 mm sle (Bett et al., 2023). Our simulations with depth-parameterized melt rates do allow the melt to increase as the ice shelf area expands though not necessarily in a way that realistically accounts for ocean circulation. The most aggressive melt parameterization (see 160_700 in Figure S2) for PIG yielded an average melt rate of 182 Gt $yr^{-1}$. If we take the corresponding VAF loss as an upper bound, then the maximum 2-century melt-driven VAF loss from PIG is 63
mm sle (Figure 10), 24 mm sle of which occurs over the first century of the simulation.

The evolution of the Thwaites cavity is more complex because, in advanced stages of grounding-line retreat, it broadens and deepens, providing a much greater area exposed to high melt rates at depth. Based on an ocean model with warm conditions used by Bett et al. (2023), melt increases from ~46 Gt/yr for the current geometry to ~220 Gt $yr^{-1}$ when the VAF loss reaches
~40 mm sle. Although their model loses mass much faster than ours due to its treatment of the ice dynamics, the melt rates the ocean model produces at a particular VAF loss should not depend heavily on the time it took to reach that state if the temperature forcing is steady as in this case. Assuming a linear increase, these rates imply an average melt rate of ~133 Gt $yr^{-1}$. For comparison, our 125 Gt $yr^{-1}$ melt rate produces a VAF loss of 39 mm sle. Thus, for the warm conditions they used, our



simulation suggests losses of ~40 mm sle over the next 200 years. As the Thwaites cavity increases in response to greater
losses, the melt rates could eventually reach 600 Gt yr$^{-1}$ (Bett et al., 2023), indicating much larger losses may be likely in the
23$^{rd}$ Century and beyond. For comparison, the most aggressive parameterized melt rate function for Thwaites (see B&G in
Figure S2) produces an average melt rate of 151 Gt yr$^{-1}$ ($h_T$=41 m), which yields a VAF loss of 46 mm sle (Figure 10). Thus,
Thwaites melt-driven losses are likely to remain relatively moderate (< 50 mm sle) over the next two centuries with our
preferred value of $h_T$ (41 m), which is comparable to PIG. After 200 years, however, as melt rates increase, losses should
accelerate rapidly (Joughin et al., 2014). If it turns out that the larger values of $h_T$ should be used in place of our preferred
value, then the period of rapid losses for Thwaites could occur earlier and could greatly exceed those from PIG (Figure 10).

## 5. Conclusions

We have conducted several numerical simulations for Pine Island and Thwaites glaciers to understand how their projected
contributions to sea level over the next two centuries are affected by the amount of melt and the choice of friction law. For
most cases, the choice of friction law makes little difference if the same loss of basal traction occurs as the region near the
grounding line approaches flotation. Our preferred value ($h_T$=41 m) produces ~20% more loss than the cases when there is an
abrupt transition from full friction to no friction as the ice column goes afloat ($h_T$ =1 m). The value of $h_T$ =41 m is roughly
consistent with the degree of weakening introduced by other regularized Coulomb friction laws. Our results indicate, however,
that the weakening itself introduced by these friction laws is a far more significant effect than the actual transition from
Weertman to Coulomb plastic conditions over a small fraction of the domain. The possibility remains that sea level
contributions could be much larger (>2x) if a value of $h_T$ substantially larger than our preferred value is found to be more
appropriate.

Our results indicate that irrespective of the choice of $h_T$, losses are a linear function of the total melt averaged over the full
simulation period. This linearity holds for simulations with both constant melt and freely evolving depth-parameterized melt.
The spatial distribution of the melt has little effect on overall VAF loss. Each glacier, however, has a different sensitivity to
melt. With its more well-buttressed ice shelf, PIG yields about 50% more VAF loss for each incremental increase in the melt
than does Thwaites. Thus, despite the complexity of the non-linear system, 200-year simulated losses from the glaciers are
reliably predicted solely by the spatiotemporally-averaged melt rate.

While we can't account for other factors that might increase ice loss, such as full ice shelf breakup (MacAyeal et al., 2003) or
partial shelf loss (Joughin et al., 2021a), our results suggest some bounds on melt-driven losses from PIG and Thwaites over
the next two centuries likely will not exceed 10 cm. Two centuries out, however, both glaciers will have lost a substantial
amount of ice and will be primed for much more rapid loss if melt rates don't subside.



**Code and data availability**

The original BedMachine bed and surface elevations and thickness data (DOI: 10.5067/FPSU0V1MWUB6) MeASUREs Phase-Based Antarctica Ice Velocity Map V001 (DOI: 10.5067/9T4EPQXTJYW9) is available at NSIDC. To allow for any updates that may occur during the revision process, we defer the permanent archiving of all other data used to constrain the model until final acceptance. Prior to final publication, all data will be made freely available with a permanent DOI at the University of Washington ResearchWorks Archive. In the interim, the data will be made available on request.

Icepack is available at https://icepack.github.io (comit 0c17259979b1e595fdfcccb53bdc6f3d033755c4). The basin-scale model and supporting code are available at https://github.com/fastice/icesheetModels (comit 5c9406443ec34f7dd857fa6265a951b9aa634b84)

**Author contributions**

IJ conceived the ideas with support from PD. The basin scale model was written by IJ and he performed the model runs and analysis of the results with input from PD and DS. DS wrote the icepack modelling package and provided guidance on its use. All authors contributed to the production of the final manuscript.

**Acknowledgments**

Ian Joughin and Daniel Shapero were funded by the National Aeronautics and Space Administration (NASA) and the National Science Foundation (NSF) and PD was funded by the National Environment Research Council (NERC).

We acknowledge data contributions from the EU/ESA Copernicus Sentinel 1A/B missions, M. Morglighem (BedMachine), I. Howat (REMA data included with BedMachine), E. Rignot (MEaSUREs Velocities), and B. Medley (SMB Map). Balance velocities were produced using code supplied J Bamber and developed by A. LeBrocq, which was constrained by RACMO 2.3 data produced by the Institute for Marine and Atmospheric Research Utrecht (IMAU).

**Financial Support**

The work at the University of Washington was supported by NASA Grant 80NSSC20K0954 and by NSF Grant OAC-1835321 and the work at the British Antarctic Survey was funded by NERC.

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







**Figure 1. Basal shear stress ($\tau_b$) as a function of speed for conditions representative of flow a) near the grounding line, b) on the main trunk farther inland, and c) slow inland tributary flow. In each case, the sliding coefficients have been selected to produce the same speed basal resistance at a nominal reference speed.**





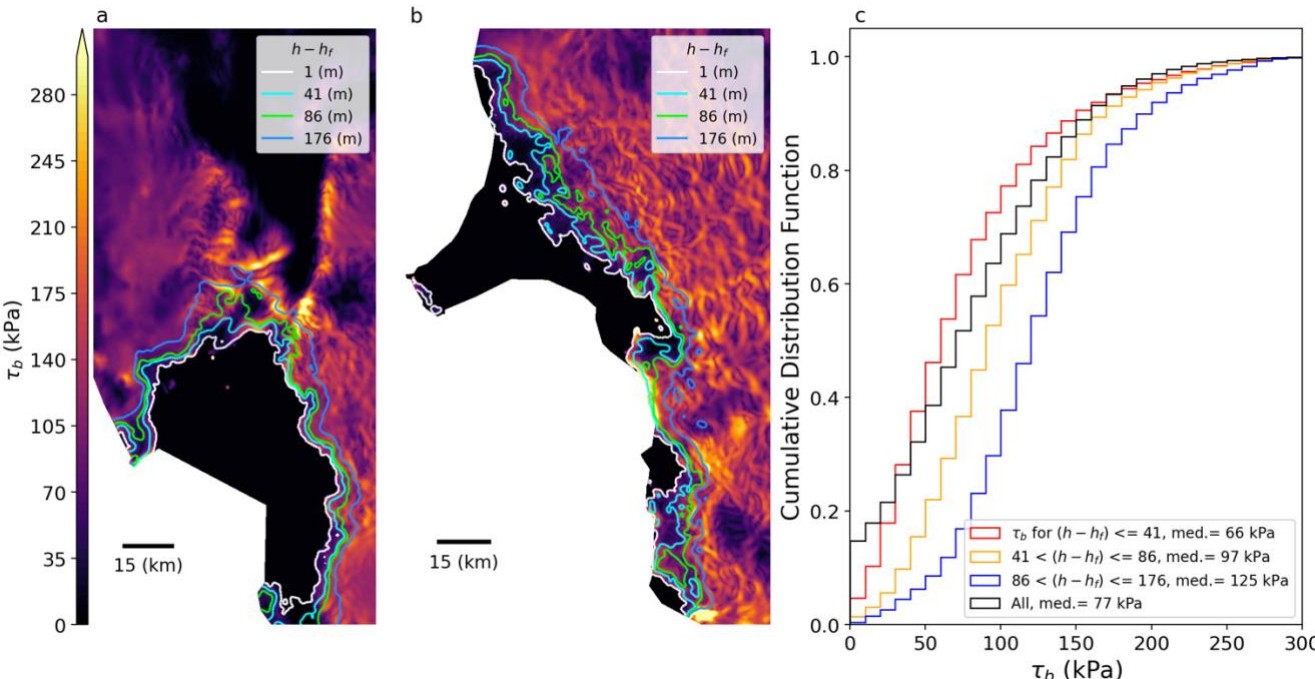

**Figure 2. Height above flotation ($h - h_f$) contours plotted over inferred basal shear stress ($\tau_b$) for (a) Pine Island and (b) Thwaites glaciers. (c) Cumulative distribution functions for $\tau_b$ for bands defined by $h - h_f$.**



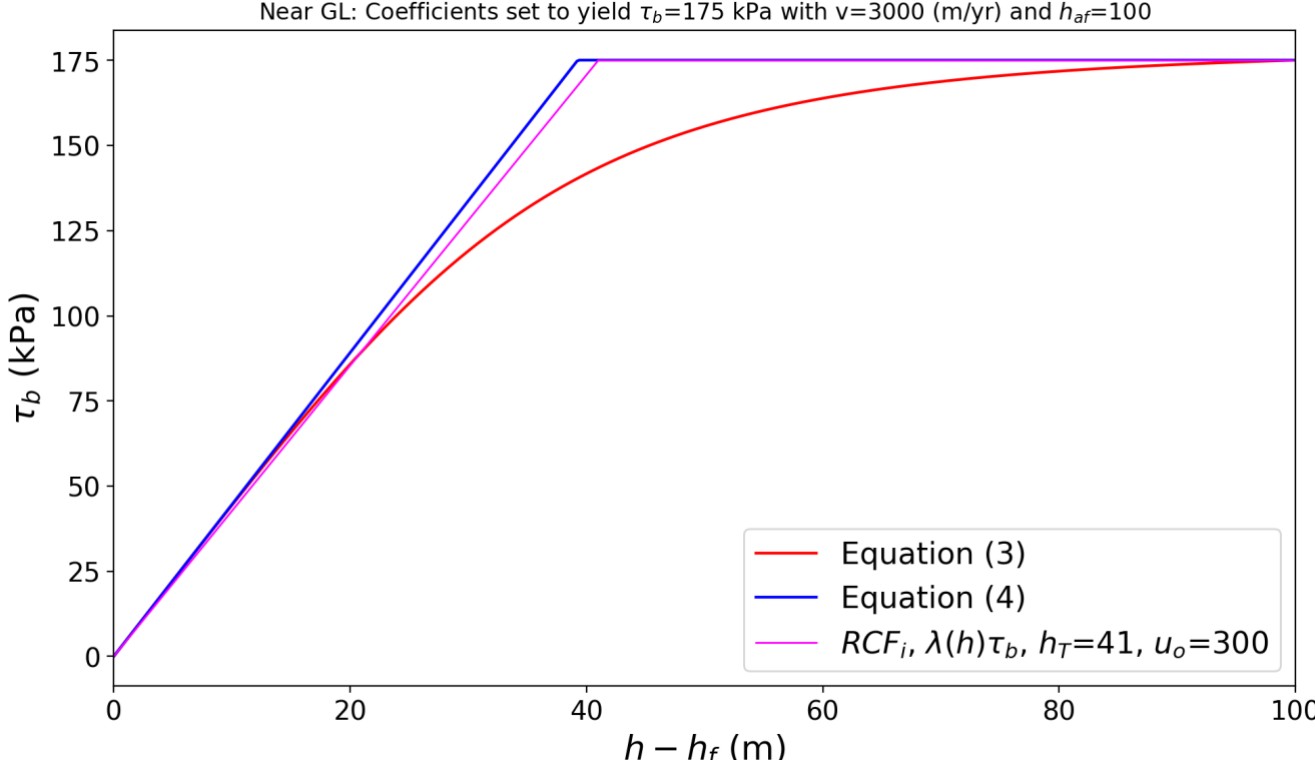


**Figure 3. Example of the decrease in basal resistance as ice approaches flotation for the different friction laws described in the text.**





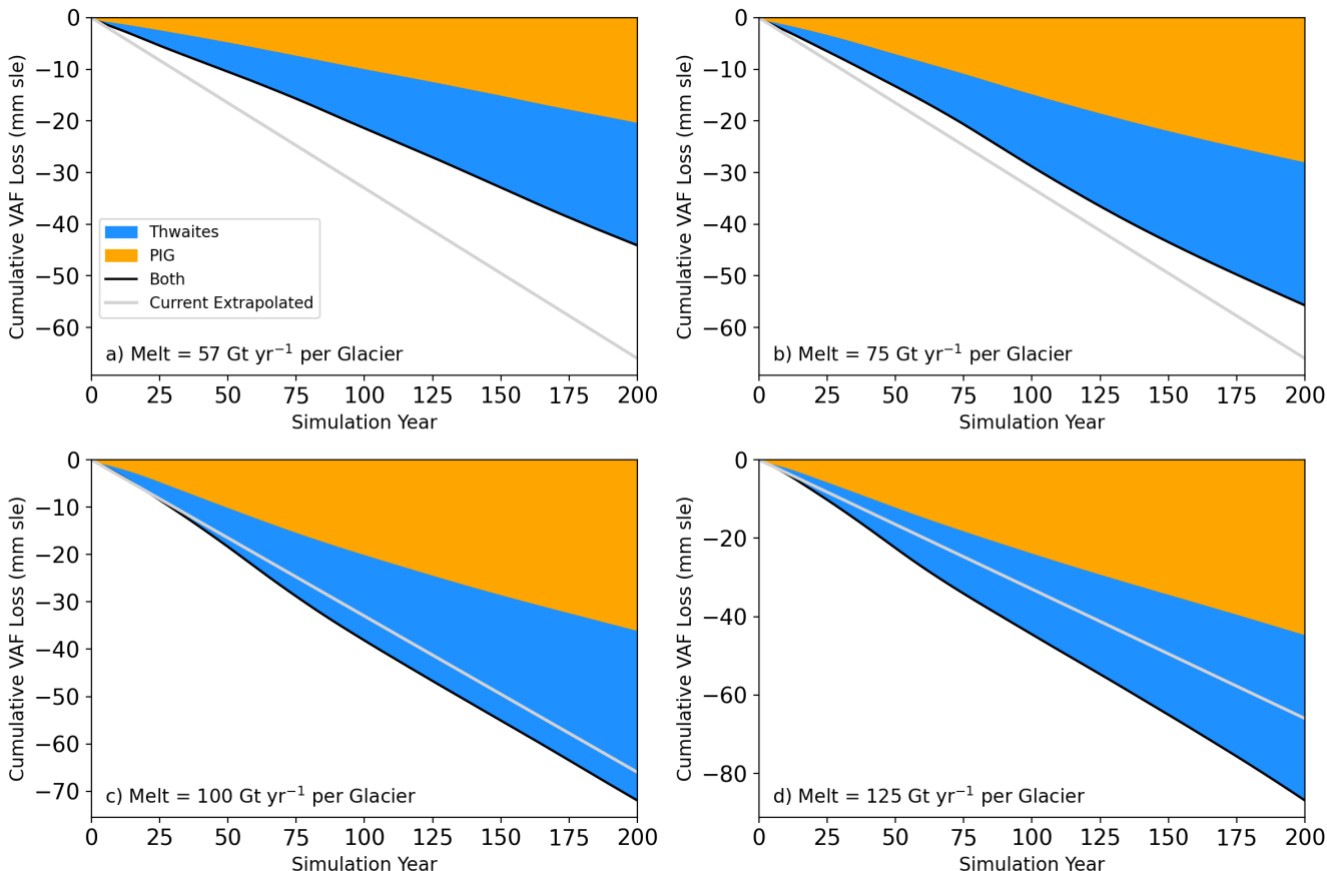

**Figure 4. Cumulative volume above flotation (VAF) losses for Pine Island and Thwaites glaciers simulated using RCFi for melt rates applied to each glacier's ice shelf of a) 57 Gt yr$^{-1}$, b) 75 Gt yr$^{-1}$, c) 100 Gt yr$^{-1}$, and d) 125 Gt yr$^{-1}$. All simulations $h_T$=41 m. Also shown is the initial combined VAF loss rate based on observed velocity linear extrapolated over 200 years. All losses are subject to map-projection-related biases of a few percent.**



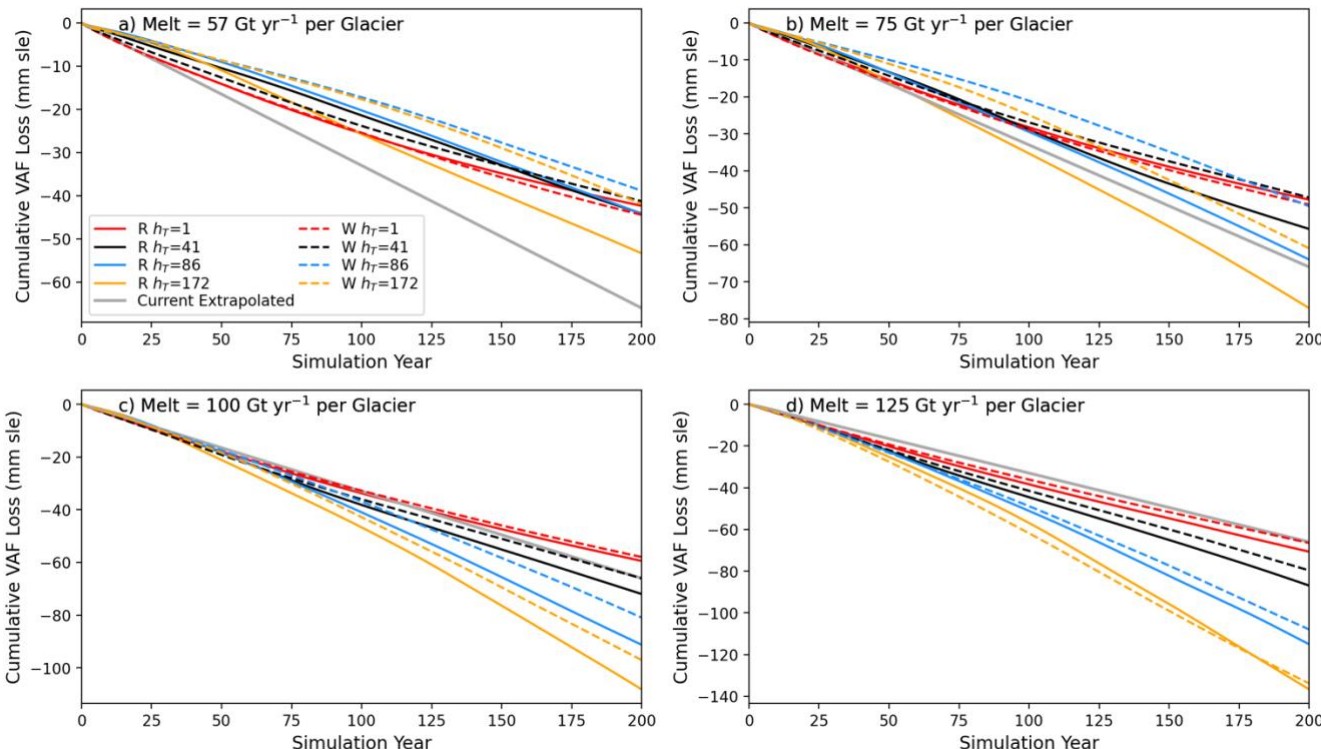

**Figure 5. Cumulative VAF loss simulated using both RCFi and Weertman sliding for several values of $h_T$ for melt applied to each glacier's ice shelf of a) 57 Gt yr$^{-1}$, b) 75 Gt yr$^{-1}$, c) 100 Gt yr$^{-1}$, and d) 125 Gt yr$^{-1}$. All losses are subject to map-projection-related biases of a few percent.**



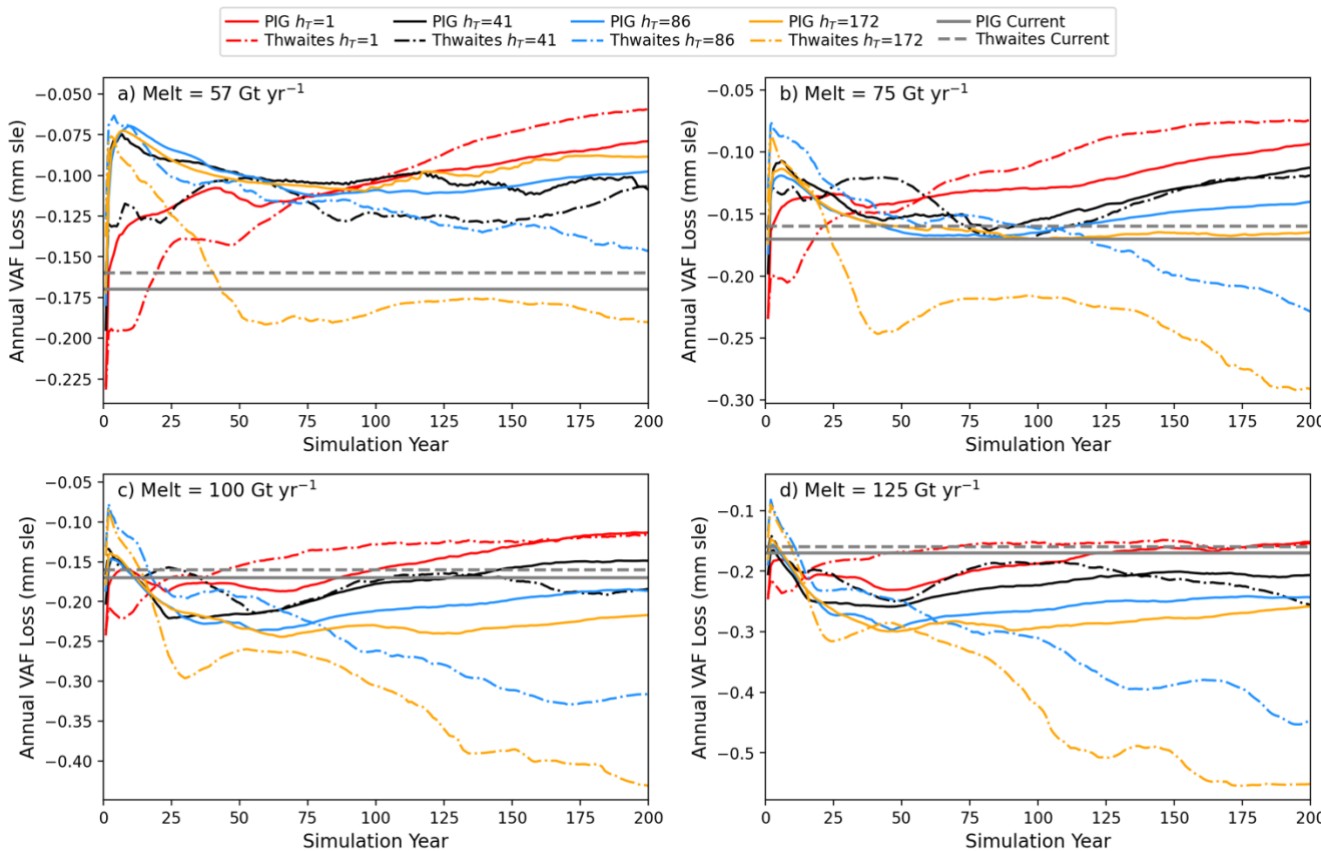

**Figure 6. Annual VAF loss for PIG and Thwaites glaciers simulated using RCFi with melt applied to each glacier's ice shelf of a) 57 Gt yr⁻¹, b) 75 Gt yr⁻¹, c) 100 Gt yr⁻¹, and d) 125 Gt yr⁻¹. The current rates based on velocities used to initialize the model are also shown. All losses are subject to map-projection-related biases of a few percent.**



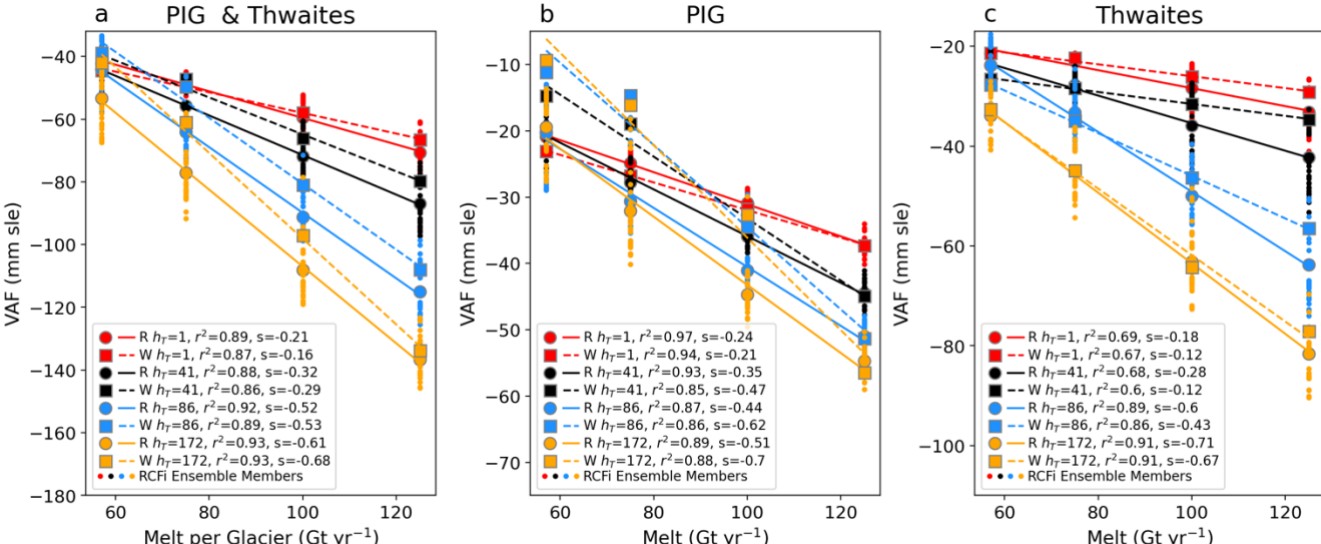

**Figure 7. Simulated 200-year VAF losses as a function of melt for a) both glaciers, b) PIG, and c) Thwaites Glacier. Results are shown for several values of $h_T$ and both RCFi and Weertman sliding. Each result represents the average of an ensemble of 30 simulations with randomly generated melt distributions, which are shown only for the RCFi simulations. The lines show linear regressions to the ensemble data (4x30 points), with the corresponding slopes given in each legend. As such, the $r^2$ values in the legend represent the proportion of the variance caused by the melt forcing.**



**Figure 8. Simulated 200-year VAF loss/gain for $h_T$=41 m averaged over 30 ensemble members for RCFi with melt applied to each glacier of a) 57 Gt/yr, b) 75 Gt/yr, c) 100 Gt/yr, and d) 125 Gt/yr, and for Weertman sliding with melt of e) 57 Gt/yr, f) 75 Gt/yr, g) 100 Gt/yr, and h) 125 Gt/yr. Speed at 1000 m/yr intervals are shown in black and basis boundaries are shown in blue. The box in panel a indicates the area shown in more detail in Figure 10.**



**Figure 9. The number of ensemble members (colour) at each point that are floating after 200 years simulated with RCFi for melt per glacier of a) 57 Gt/yr, b) 75 Gt/yr, c) 100 Gt/yr, and d) 125 Gt/yr, and with Weertman sliding for melt of e) 57 Gt/yr, f) 75 Gt/yr, g) 100 Gt/yr, and h) 125 Gt/yr. The speed at 1000 m/yr intervals is shown in black.**



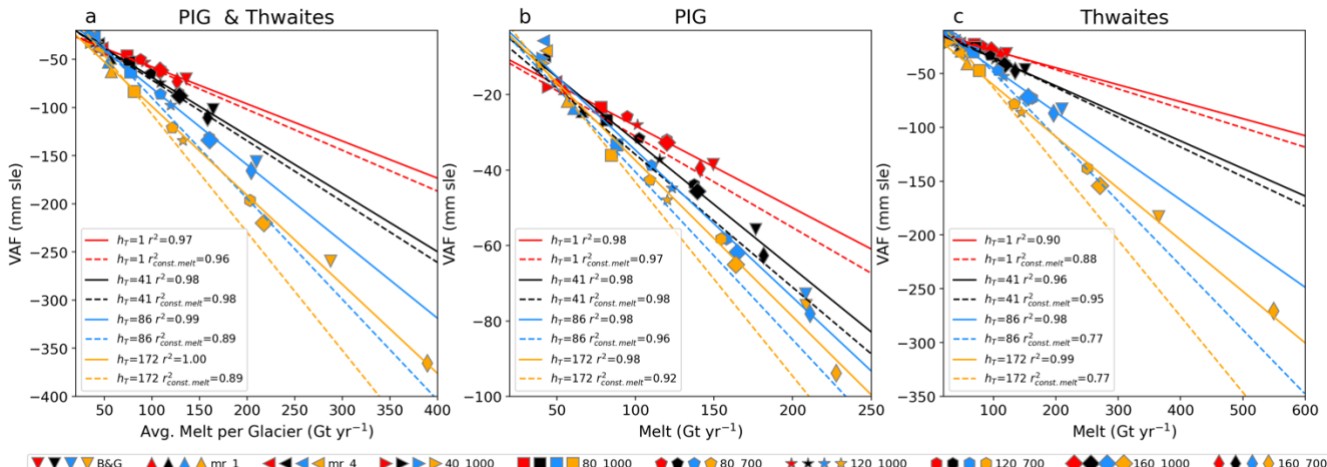

**Figure 10. Simulated 200-year VAF losses as a function using depth-parameterized melt functions for a) both glaciers, b) PIG, and c) Thwaites Glacier with RCFi. Results are shown for several values of $h_T$. The solid lines show the linear regressions to the plotted points, and the dashed lines are those computed from the ensemble data shown in Figure 7.**