# Peer review of "Responses of Pine Island and Thwaites Glaciers to Melt and Sliding Parameterizations"

_EGUsphere, 2023_

## Referee Comment (RC1)

**Title**: Response of Pine Island and Thwaites Glaciers to Melt and Sliding Parameterizations
**Authors**: Joughin, I., Shapero, D., and Dutrieux, P.
**Journal**: *The-Cryosphere*
**Reviewer**: Tyler Pelle (Scripps Institution of Oceanography; tpelle@ucsd.edu)

**Overview**:
In this work, Joughin et al. investigate the impact of using both Weertman-style and regularied Coulomb friction law, in conjunction with a linear scaling to the associated basal shear stress field for each law that enhances bed weakening with proximity to the grounding line, on 200-year simulations of Pine Island and Thwaites Glaciers. In addition, this paper also compares the response of these glaciers to an ensemble of randomly generated ice shelf basal melt volumes. The authors find that the choice of friction law has a relatively minor impact on ice volume changes of this sector and recommend use of a regularized Coulomb friction law when only one friction type is used. In addition, parameterized bed weakening led to significant enhancements in the 200-year global sea level contribution of this region, highlighting that such weakening should be included in future ice sheet model simulations either through this explicit manor, or through an effective pressure dependance. Lastly, the sea level response of the system was found to have a strong linear dependance on the total integrated ice shelf melt volume through the 200-year simulations.

I have to admit that I had a bit of a tough time working my way into this paper because the basal friction overview is quite long and involved, which I think might be a bit much and could be simplified to only include information that is needed to support the results/discussion/conclusions. I also found that visualizing differences in some of the figures was challenging because of the use of different y-axes limits, many of which I think can be made consistent. However, once into the results, I found this work to be an absolute pleasure to read and is full of really wonderful conclusions and insights that would be of wide interest to both the ice sheet modeling and broader scientific communities. It is also clear that this manuscript has been built on a long line of research that the authors have been working on for quite some time, so it is great to see everything come together in such a wonderful way. Below, I provide a number of suggested edits the authors can make to improve the manuscript, most of which aside from restructuring are small-technical corrections that should be easy for the authors to address. Due to the restructuring of the beginning, I suggest major revisions; however, I am very supportive of publication once these comments have been addressed.

**General Comments**:
- **Basal Friction Overview section**: This section is quite long and I have to admit that I got a bit confused reading through it, which was challenging because it sets the stage for the rest of the paper. There's a lot of analysis and equations presented and I am wondering if it is possible to shorten this section to only what is necessary for interpretation of

results/conclusions? After reading through the paper, I think the critical information needed in this section is that *a linear scaling that depends on a height above flotation threshold $h_{T}$ (found in previous work to be ~41-46 m for PIG) is applied to $\tau_{b}$, which is solved for via both RCF and Weertman-style friction law. In this paper, we will investigate how variation in $h_{T}$ applied to both the BCF and Weertman sliding laws, as well as ice shelf melt, impacts future ice loss of PIG and THW.*

Given that, I think both friction laws that are used in this paper, as well as the linear scaling, should be provided, and other information should be limited as needed for clarity. For example, there are a few forms of Weertman sliding laws presented (eqn. 1, 6) and I am not sure which one is used in the simulations. Also, the friction law that combines Coulomb and Weertman (eqn. 4) is not used in the rest of the paper, but stimulates quite a long discussion about $h_{f}$ transition points (L81-89) and effective pressure (L91-109; which is not used in either of the friction laws used in this study since it is subsumed into the friction coefficient solution). While I think this information is really fascinating and I think the authors did a great job on the analysis, I unfortunately don't think it is appropriate in its current place in the manuscript and suggest the authors revise this section and shorten it considerably. Perhaps a lot of this can go into an appendix, with information in the main manuscript saved for only what is most pertinent?

- **Consistency of language and figure axes in manuscript**: I noticed many different forms of "PIG and Thwaites Glaciers", "PIG and Thwaites glaciers", and "Pine Island and Thwaites glaciers". In my opinion, I feel like the "PIG and Thwaites Glacier" would be the most correct, but I am very happy for the authors to choose their favorite variation and use only it throughout the main manuscript, supplement, and in figures and associated captions. I pointed out a few places in the manuscript where I noticed this, but I likely did not catch them all, which is why I raised it as a general issue. On a similar note, I also noticed many variations of y-axes limits on figure panels that could be made consistent across the entire figure. This could help visualize differences in figures where there are many intersecting lines and many panels.

**Line Comments**:
- L8: Specify ice shelf basal melt here so readers know you are not referring to melt of grounded ice.
- L11: Change "above" tp "upstream of"
- L14: remove "work" – i.e. "our simulations suggest"
- L26: "continued melt forcing" – I think you are referring to ocean induced ice shelf basal melt based on the citations, but I think it would be helpful for readers if you were explicit about this here. Perhaps here you can say " . . . continued ocean forcing in the form of ice shelf basal melting (hereon referred to as melt)."
- L33-34: I think it would improve readability if the definition of variables ($\tau_{b}$ and $u_{b}$) is confined to the "Basal Friction Overview" section.

- L40: By models, do you mean basal friction parameterizations? If so, please specify because you use "models" twice in this sentence that means two different things.
- L50-54: As per line comment L33-34, I think this would be a perfect place to introduce variables $\tau_{b}$ and $u_{b}$.
- L51: Is this the Weertman sliding law used in the simulations, or is it eqn. 6?
- L70-90: I think there is a lot of interesting information here but it is quite lengthy. If the main point is that we expect Coulomb sliding behavior near the GL and at some transition point, we expect Weertman sliding, maybe this can be said more succinctly with less analysis? This seems very in-depth for a section that is not either the results or discussion.
- L81-89: I have to admit that I got a bit lost with some of the analysis here. In particular, I am a little confused about where the value of 45 m came from and why this is only computed for the near GL region (Fig. 1a) - can a similar value be computed and added to the plots for trunk (Fig. 1b) and inland tributary (Fig. 1b)? Also, in figure 2, are you plotting $h-h_{f}$ as per the equation at the end of line-83? If so, this should be mentioned in this paragraph and also in the figure caption of figure 2. For figure 2, why are the values of $h-h_{f}$ (1, 41, 86, 176) chosen?
- L93-95: There are recent modeling efforts that model upstream Thwaites/PIG effective pressure with subglacial hydrology models (e.g., Hager et al., 2022; Dow et al. 2023) that show low effective pressure far upstream of the grounding line - I would recommend citing them here since they support your claim. It is difficult to know how accurate these model simulations are, but they are likely the best we can do at this point!
- L129: "Close to the grounding line" is defined as $h-h_{f}<41$, but figure-2 shows that there are numerous regions where the contour of $h-h_{f}=86$ is nearly superimposed onto the line $h-h_{f}=41$; why was the value of 41 m chosen?
- L135: When you say "as the surface elevation approaches flotation", are you referring to in figure 3 when $h-h_{f}$ approaches 0, meaning when $h=h_{f}$? In line 78, you defined h as the ice thickness, so is surface elevation accurate? Perhaps just saying "as ice approaches flotation . . ." would be more clear?
- L140-145: While I think it is obvious in eqn. 8, I think it should be reiterated that this linear transformation applied to $\tau_{b}$ evolves in time to the changing ice thickness in your simulations.
- L161: You have two section-2's (the friction overview and methods section).
- L177: Need a new paragraph space between these paragraphs
- L178: Do you mean Equation 7?
- L186: I think it is worth mentioning that your computed fields for the friction coefficient and A do not change in time.
- L197: Is the ice front fixed (i.e. no calving is simulated)? Is this a major limitation given that your 2021a work indicated that ice shelf retreat was the primary driver of recent

speedup of Pine Island Glacier? If so, I think this should be mentioned somewhere in the text.

- L199: I'm just noticing this now, but I think it would be more correct to say "PIG and Thwaites Glacier" since PIG stands for Pine Island Glacier. See general note about consistency through manuscript.
- L208-215: This compiled velocity map is very interesting, it would be great to see a figure of it somewhere in this paper! Perhaps the same can be done for the SMB and combined into one figure?
- L220: Does SMB vary in time in the simulations?
- L240-245: Another interesting pattern in figure-5 is that at higher melt rates (panels c/d), corresponding Weertman and RCFi mass loss time-series seem to align more than for lower melt simulations (in the top two panels, the Weertman lines are all clumped above the RCFi lines). That is, it seems like the choice of sliding parameterization becomes less important when the system is forced with higher melt rates. Do you agree? This could be an interesting point for the discussion.
- L250: Please change "Thwaites" to "Thwaites Glacier" here and also through the manuscript. Also, there are times when you capitalize "Glacier", and when you keep it lowercase (i.e. Thwaites glacier, L261) - I think it is correct for it to be capitalized, so please make that consistent throughout the paper as well.
- L263: Change ". . . sensitivity for PIG is -0.24 to -0.51 mm $Gt^{-1}$yr sle for PIG and . . ." to ". . . sensitivity is -0.24 to -0.51 mm $Gt^{-1}$yr for PIG and . . ." (you said "For PIG" twice here)
- L270: Is the unit supposed to be "a few hundred m/yr" or "a few hundred m", which seems more in line with the values in figure-8.
- L277: Change "above the grounding line" to "upstream of the grounding line" here and throughout the manuscript.
- L281: The notation "Figure 9e&f" seems a little messy, perhaps could use "Figure e/f"? I think this is personal preference and maybe a little pedantic, but I have not seen the "&" symbol used in this manor in a manuscript before.
- L287-289: It would be nice to see what some of these melt distributions look like since they seem like a primary control on the spatial distribution of retreat of Thwaites Glacier.
- L317-318: I would even say that simulations with greater $h_{T}$ values lose less mass than those with higher $h_{T}$ values for these PIG simulations, which is pretty fascinating! I really enjoy your call to earlier work here that investigated this paradox.
- L346: I'm not sure if it is just the way my computer rendered the PDF, but it seems like there is an unnecessary space in the word "ri ght", but I'm not sure if this is a typo.
- L350-357: I really enjoy this discussion and am excited with the prospect of future work that compares such implementations of bed weakening near the grounding line.
- L359-374: I understand the reduction of section "Basal Friction Overview" might cause issues with this and the preceding discussion paragraph, but I think if you put a lot of the

"Basal Friction Overview" information in an appendix, you can still keep these paragraphs (with references to the equations in the appendix), which I think are very important findings and thoughts from your work.

- L377: I've noticed a few variations of "PIG and Thwaites Glacier" (L377), "PIG and Thwaites glaciers" (L261), and "Pine Island and Thwaites glaciers" (L770). Please pick one and keep consistent throughout the manuscript, supplement, and all figures and figure captions.
- L406-417: See note for figure 10 below - In short, I would recommend a different way of showing these results. Given that this paragraph focuses mainly on the associated $r^{2}$ values, I'm wondering if figure 10 could be consolidated into a table?
- L427-429:I got confused by the phrasing "results lie along a line" and "results that fall well off a line"; it took me a second to realize you are referring to a linear regression line. Maybe rephrase to clarify.
- L430-432: Seroussi et al. (2023) found that treatment of ice dynamics was the main driver of uncertainty in the ISMIP6 ensemble through 2100; however, this was across the entire AIS. In line-431, maybe specify that you are only referring to the ASE (i.e. ". . . suggests that differences between models in this sector may largely . . .").
- L440: This is true for the ASE, but would these conclusions hold for other sectors of Antarctica? If you are not sure, it might be worth specifying here that you are referring to coupled ice-ocean models of the ASE.
- L502: Consider rewording: " . . . our results suggest that melt-driven losses from PIG and Thwaites Glacier over the next two centuries likely will not exceed 10 cm."
- L511-512: Please remove hyperlinks – also, I think "comit" should be changed to "commit" in this section.
- L797: Magenta box in panel-a denotes domain for figure 9 (not figure 10)

**Figure Comments**:
- It can be quite challenging to compare VAF losses between panels in figures 4-7 since the y-axis limits are all different. Where appropriate, can you please use the same y-axis limits? I think this would be very helpful for all panels in figures 4-6, as well as panels B/C of figure 7. Otherwise, all of the curves look fairly similar and it can be difficult to visualize differences between them, which I think is the ultimate goal of these figures.
- Figure 2: Perhaps it was said in the main text, but it would be good (in the figure caption) to reiterate the friction law that was used to compute the basal shear stress that is shown in panels-A/B.
- Figure 3: The text and lines in this figure are a little blurry and/or choppy (as with figure 1 as well). Will you output these as PDF's in the final submission?
- Figure 8: It doesn't seem like two color bars are needed, but rather you could use one that diverges at 0, with blue (negative) trailing off to the left, and red (positive) trailing off to the right. Also, would it be possible to scale the intensity of the blue so that it uses the

same color scale as red (i.e. reaches maximum intensity at -500 m). This should result in very pale shading of blue where you have height above flotation gains, which would seem more appropriate since now, it appears like the gains in the bottom and upstream parts of the domain are stronger than the losses Thwaites Glacier experiences, which is not true.

- Figure 10: I believe I understand that the main point of this figure is to show that the linear relationship between total integrated melt volume and VAF change holds for various other melt parameterizations, but I think the number of different-shaped symbols is very distracting and is rather uninterpretable for the reader. Many of the symbols are clumped over each other at lower x-values and also, the meaning of the different symbols is not explained in the main text (i.e., I know the different symbols are different melt function outputs, but I have no idea what "mr_1", "mr_4", "80_700", etc. are specifically), and I don't think readers should have to read the supplement to interpret a main-text figure. Ultimately, I don't think the symbols matter given that the analysis is not dependent on individual melt functions, so I wonder if the authors can remove the 10 symbol-types and use just one symbol and the different colors to represent the different melt function outputs? I per line comment L406-417, I think this figure could also be consolidated into a table given that the main piece of information that is used from it are the $r^{2}$ values.
- Figure S1: No change is needed in the text, but I am just curious what metric you use to prescribe your mesh resolution?
- Figure S2: In the figure caption, please include the full in-text citation for Barnes and Gudmundsson (2022).

---

## Author Comment (AC1)

**Response to Reviewer 2**

Joughin et al. explored the sensitivity of projected 200-year mass loss from Pine Island and Thwaites Glaciers to the friction model, weakening of basal drag upstream the grounding line when ice approaches floatation, and the sub-shelf melt rates. They find relatively small differences between Weertman and Coulomb sliding laws but high sensitivity to the rate at which the basal drag is reduced when the ice approaches floatation. They also find the sea level contributions from both PIG and Thwaites glaciers are less sensitive to the spatial or temporal distribution of melt. The sea level contributions from these two glaciers is not likely to exceed 10 cm before end of 2200. Overall, the manuscript is well structured. However, the model setup section and some of the descriptions need improvements. There are quite a few typos in the texts and figures, especially the numbering of equations, which are misleading and need to be fixed. Information of some captions are insufficient and some statements in the discussion are a bit far-fetched.

*Summary of more specific comments that are addressed below.*

**Here are some general comments:**

Some of the methods description is not quite clear. I suggest more details in the way how you generate the basal melt distributions at four different levels.

*We provided a high-level summary of the melt procedure along with a reference to the original paper where the melt model is described in more detail. We did not see the need to repeat the detailed description here and refer the reviewer to the reference. With respect to the 4 levels of melt, the original text explicitly says "At each time step, each melt distribution is re-normalized to produce a specified level of melt (e.g., 57 Gt/yr)." In response to this comment, we added the "(e.g. 57 Gt/yr)" to make this point more clear.*

Is it also not clear how you treat the basal drag and basal melt at the grounding line.

*Regarding basal stress, as the submitted manuscript states "Here we perform experiments using Equation (8) because it allows us to vary the amount of weakening so that we can study the resulting impact on ice loss." Equation 8 forces the basal shear stress to 0 at the GL (i.e., as the height above flotation goes to zero).*

*To make the melt treatment clearer, we added the underlined text to the existing text. "For most of the experiments, we use a randomly generated ensemble of 30 melt distributions applied to the floating nodes (Joughin et al., 2021b).*

One of the key findings here is the small difference between Weertman and Coulomb friction model. However, this statement has a very important premise that the

authors applied a linear weakening of the basal drag for both sliding laws. The statement will not hold without this premise, which should be mentioned in both abstract and the conclusion.

> *Coulomb friction and Weertman sliding are both friction laws that can be treated as a function of the effective pressure. If the effective pressure is well known then both should include the effective pressure dependence. Lacking such knowledge, a wide variety of treatments have been used by the modeling community ranging from completely dropping the dependence (Equations 1&2) to using an assumed value for the effective pressure  (Equations 3, 4, and 5). And there is no reason that changes in effective pressure near the grounding line should not introduce similar weakening as for the Coulomb case (again when not simplified, both have a dependence on N – see Eq. 6 ).*
>
> *A major focus of this paper is to try to systematically separate the effective pressure response (near grounding line weakening) from the friction type (Coulomb versus Weertman). So, we have pulled things apart in the way that we have to accomplish this objective. Readers can examine the results and draw their own conclusions.*
>
> *The lead author has published more than half a dozen studies using this weakening approach. It works well for us. We do not, however, make the point this is the only way or best way to model the weakening. In fact, we conclude that both laws can be parameterized to produce similar results. "The fact that our empirically-derived value of $h_T$ agrees well with roughly equivalent values determined from consideration of effective pressure suggests that both types of models tend to reduce basal traction at rates that are approximately the right magnitude."*

 Figure 7 and 8 even show a clear difference under low melt cases between Weertman and RCFi, which was not well discussed in the paper. In Fig 8, the mass loss at the front of the Thwaites Glacier is clearly higher in Weertman than using RCFi.

> *We reconsidered and rewrote the explanation to say "Overall, our results indicate the choice of friction law yields relatively minor differences to the projected VAF losses (Figure 7), except for the PIG cases with low melt. These differences are consistent with the PIG re-grounding seen in the low melt simulations with Weertman sliding (Figure 9e&f). As noted above, there are limited areas ($h_o < h_T$) where the bed can strengthen if thickening rather than thinning occurs. Such thickening rarely occurs because the region near the grounding line tends to nearly always thin. For some Weertman cases, however, thickening and advance do occur for sufficiently low melt, which should be reinforced by thickening-induced strengthening of the bed near the grounding line. This would explain why the losses decline as $h_T$ increases for the low melt Weertman cases on PIG, since the area subject to this type of strengthening*

*expands. Whether this should remain a feature of our model is a subject for future research."*

This study found the ice mass losses are highly sensitive to the choice of $h_T$. However it's not clear about how to choose the best value of $h_T$ if we change to a different glacier? The best fit of $h_T$ = 41 m for the Pine Island Glacier may not be the best choice for other glaciers in Antarctic or even for the Whole Antarctica. $h_T$ may need to be adjusted based on the significant geometry change with time. This should be more discussed in the paper. Moreover, the study suggests a spatial variable Ht is feasible in future study. Then how about the changes of $h_T$ with time? In Joughin 2019, a 20 years simulation suggests that 46 m is the best choice. However, in a century scale simulation, the geometry near the updated GL may change a lot, which is not discussed yet.

*The same can be said of all sorts of parameters used in ice sheet models. For example, an appropriate value of $\alpha^2$ must be selected and could also be glacier-dependent and time-varying. There are few if any model results that let the drag coefficient $\beta^2$. It's not an ideal situation, but making such assumptions is often the only way forward.*

*While we have only empirically evaluated our choice based on the time-dependent response of PIG that is a more quantitative evaluation of the parameter than often is the case. A major point of the study is to evaluate the sensitivity to the choice of this parameter. To make this point clear we added.*

*"Similarly, our best estimate for $h_T$ is based solely on the response of PIG over a decade and half. While it is likely that other glaciers can be modelled well with a value of $h_T$ of similar magnitude, further work is needed to establish the best value for other regions. Our results, however, do establish that choice of $h_T$ can have a substantial effect on projected losses as is the case for $\alpha^2$ (Barnes 2022)."*

The last paragraph of Sect 4.2 is discussing the shortcomings of other sliding laws and conclude that 'any law that relies solely on the local height above flotation to govern changes in effective pressure, and thus, basal friction over the entire domain is likely oversimplified and incorrect'. However, the authors did not discuss the shortcomings of RCFi in this study, like how to better decide $h_T$ used in RCFi considering different glacier may have different sensitivity to the choice of $h_T$ as you show in Fig 7b and 7c for the high melt level case.

*To be clear, this comment is directed only at Budd friction, which makes this assumption for the entire domain. This should be clear from the sentence, but we added. "(.e.g., Equation 6)"*

*See comment above. In addition, we feel the strength of RCFi is that it applies Coulomb friction to areas of the bed where the ice motion is fast and where theory (see Schoof reference) and empirical work (see Zoet reference). Furthermore, RCFi has no parameter hT. That is introduced through weakening function $\lambda(h)$. We could retain RCFi and define a different weakening function with similar dependence as the other Coulomb friction laws as we clearly stated in the original submission. "There is no reason, however, that $\alpha^2$ in Equations (3) and (4) cannot be selected through a procedure like that used to derive our preferred value of $h_T$. On the other hand, Equation (8) can easily be modified to have a spatial variable $h_T$ that depends on effective pressure in a similar manner to Equations (3) and (4), which would allow the traction reduction to be decoupled from the form of the basal friction law."*

**Another statement in this paper is about the low sensitivity to the spatial or temporal distribution of melt rates. However, Fig 7and 9 did show sensitivity of GL retreat to the spatial distribution of melt as the author mentioned on Line 287-289. Recent studies indicate that the migration of the grounding line is extremely sensitive to how basal melt occurs adjacent to the grounding line (Arthern and Williams 2017; Reese et al., 2018; Goldberg et al., 2019). Modelling studies also suggest that ice sheet models are more sensitive to melt rates near the grounding line than to cavity-integrated melt rates beneath ice shelves, e.g. Gagliardini et al., 2010; Reese et al., 2018; Morlighem et al., 2021. Can you please justify it?**

*Simply put this is not something we find in our model.*

*Re Gagliardini et al., 2010 and another paper by Walker et al, 2008, we noted in the 2021 paper, "First, both earlier works used spatially dependent melt distributions rather than depth-dependent distributions. As a consequence, ice shelves in those models cannot respond by thinning to alter the melt intensity near the grounding line. Thus, high melt at a point near the grounding line can only be reduced by grounding-line retreat. By contrast, in our model and most other models, the simulated ice shelf can thin to redistribute melt without necessarily causing the grounding line to retreat. In addition, one of the earlier studies fixed the melt rate while allowing the total melt to increase as the shelf extent increased (Walker et al, 2008), so the total melt may have been an important factor for that model as in our results. The other of the two studies fixed the total melt by altering the shelf length (Gagliardini et al, 2010), which introduced an additional calving term that reduces the ice-shelf volume to a similar extent that more melt would have (Fig. 3). Last, both 1D models are fed by a fixed inflow of ice from a relatively short distance upstream of the grounding line. For PIG, however, a broad interior basin converges on the narrow main trunk, allowing inflow to the trunk to increase as the glacier speeds up. Unlike the case for models with fixed inflow, this extra inflow can moderate rates of grounding-line thinning and retreat (7)."*

*For the other references, there are several reasons for the differences. For example, Arthern et al used an order of magnitude coarser resolution than we did, which could easily explain the difference they get by applying melt to partially floating nodes. Goldberg et al get a variety of melt rates based on cavity geometry, which does not disagree with our result, which uses a fixed cavity-integrated melt rate. The Morlighem et al paper uses an automatic differentiation approach that linearizes the non-linear model. It's not clear to us that direct comparisons of that with the full evolving model is anything like apples to apples. The Reese et al model is diagnostic and does take into account the time-dependent response; the ensuing response to a change on the shelf can greatly modify the overall response.*

*Since we have already argued this point in our earlier work and provided a reference to that work, we have not made additional changes with respect to this point.*

Lastly, I think it is very important to point out that the conclusion 'our work simulations suggest that melt-driven combined sea-level rise contribution from both glaciers is unlikely to exceed 10 cm by 2200' is under some assumptions, like the $h_T$=41m is the best fit for both glaciers in the coming two centuries.

*For the largest fixed melt value, the 41 m value yields about 8 cm. Using hT=86 m, which is probably too large (we picked the larger value to examine sensitivity) the result not much over 10. So hT could still be substantially larger than 41 (almost double) and this statement would still be true. Nonetheless, it was a bit too strongly worded, especially in the abstract without the context of the discussion in the paper. We softened the statement with:*

*"Based on recent estimates of melt from other studies, our simulations suggest that melt-driven combined sea-level rise contribution from both glaciers may not exceed 10 cm by 2200, though the uncertainty in model parameters allows for larger increases. "*

Specific Comments:

L66: what is the citation for typical value of  to be 0.5?

*Changed to "(typically 0.5; e.g., Asay-Davis et al., 2016)"*

L82: Why the  is much higher near GL (175 kPa) compared with Trunk (10 kPa) and inland tributary (100 kPa)? Same for title of Figure 3.

*Because that's what the inversion for basal shear stress shows. We modified as follows (underlined text was added). "Figure 1 illustrates the sensitivity of these*

*friction laws to speed for parameters meant to represent the near-grounding line region, central trunk, and outlying tributaries of PIG (see basal shear stress map in Figure 2a)."*

*We did modify the Figure 1a caption to indicate it is referring to the strong bed upstream of the grounding line.*

**L83: You mean haf =45 m here rather than hf right? If yes, I think you need to explain what haf is.**

*Fixed by making consistent. Dropped $h_{af}$ in figures and replaced with $h - h_f$ to make it consistent with the text.*

**I don't quite follow the legend on Figure 1 and this sentence. What's the meaning of different values of haf for Eq(3) and (4)?**

*For the parameters we picked, h-hf =45 m would be identical to the Weertman curve, which is overplotted multiple times already. So we didn't include it. To make this point clear, we added the following to the caption*

*"For the near-grounding-line case, the transition to Coulomb friction begins for $h - h_f < 39.2$ m using Equation (4), so the curve for this equation with $h - h_f = 45$ m is not shown to avoid an additional overplot of the Weertman curve."*

*We did add another curver (h-hf=40) to show the full Coulomb conditions. We removed the Equation (4) curves, since the didn't add much.*

**Why do you say the Weerman condition is not fond where hf = 45?**

*Inspection of the figure will show all of the Equation (3) and (4) results for the parameters we used over plot the Weertman result. The one exception is Equation (3) for near-grounding line case where h-hf=45 m. We reworded the sentence a bit to make this point more clear. It now says:*

*"). In these examples, Weertman conditions are found everywhere except for the case where Equation (3) is plotted using a height above flotation $(h - h_f)$ of 45 m (transition to Coulomb) and 40 m (nearly full Coulomb; Figure 1a)."*

**L85: "transition to Coulomb sliding" how do you get the number of 67 m here?**

*Reworded to:*

*Thus, if we assume ~300 kPa as the maximum expected value for $\tau_b$ with Coulomb friction, then the transition to Weertman sliding takes place at locations where the elevation is less than 67 m above flotation for $\alpha^2 =0.5$.*

*The value of 67 m comes from using the transition value $h - h_f = \frac{\tau_b}{\alpha^2 \rho_i g}$, given in the sentence above using 300 kPa (this relation follows easily from equating $\tau_b = \alpha^2 N = \alpha^2 \rho_i g (h - h_f)$ and solving for $h - h_f$.) This seemed too trivial a calculation to include and Reviewer 1 is pushing us to shorten not lengthen this section.*

**L87: please refer to the section you did the inversion for rather than just saying 'as described below'.**

*Replace "below" with "in the Methods Section"*

**L86: I still don't understand how you pick the four values 1, 41, 86, 176 m here. 1 m is easy but how about the rest three values?**

*We agree the numbers are a bit odd. The 176 number has been corrected to be 172. They are meant to be doublings of 41 (but as I recall the base value was originally slightly different). At some point, these values stuck, and it was not worth the re-running the simulations (weeks of computer time) with 41, 82, and 164. While somewhat inelegant, the slight deviations in no way affect the conclusions.*

**L119: The legend did not show Eq(7) at all. I guess the light blue line for RCF equation (6) should be RCF equation (7)?**

*Good catch. At some point, the text was updated, which changed the equation numbers, and the figure was not updated. The legend has been revised 6 to 7, and 8 to 9.*

**L121: I think it is worth mentioning that Gillet-Chaulet et al., 2016 used a power law rather that Coulomb law.**

*Modified to say:*

*Another study indicates PIG conditions are reproduced better with a power law using values of m in the range of 10–20, which produces a sensitivity of $\tau_b$ to speed that more closely resembles that of Equation (7) than that of Weertman sliding (Gillet-Chaulet et al., 2016).*

**L147: In the text, you refer to Eq 7 for pink line of Fig3, but the legend says the pink line is from Eq 9.**

*We changed the legend to "$\lambda(h)\tau_b$" since the same weakening is applied with RCF, RCFi, or Weertman.*

**L154: hT = 41 or 46 m**

*We specified a range because we are citing results from multiple papers with slightly different values.*

**L155: it should be 2021b rather than 2021a. hT = 123 m, m = 3**

*We added the units and exponents and put the range of values from the two earlier studies to be consistent with the RCF results.*

*"(Weertman sliding with m=3 produced best results with $h_T$ = 122–123 m). "*

**L178: Equation (7)**

*Fixed.*

**L186-189: It's not clear about the sequence of inversion here. Do you invert basal friction law parameters first and then A with a second inversion or invert both at the same time? Which sliding law do you use in the inversion?**

*The inversion iterations for the drag coefficient and A are interleaved. More details are in our earlier paper. We added a reference to Joughin et al, 2021b.*

*We amended as follows to make clear we inverted for the friction law used in the corresponding forward model (i.e., Weertman inversion for Weertman simulation).*

*We initialize the model by inverting for the basal friction law parameters ($\beta^2$ for Weertman or RCFi as appropriate)....*

**L193: It is not clear about how you generate the melt distribution until I further read through the whole text. I suggest you specify how you treat the melt distribution here. I suspect you run each experiment with 30 melt distributions and normalise it to four different melt levels (57, 75, 100, 125 Gt/yr), and then update this melt distribution with an updated grounding line position. What is the time step size?**

*There is more detail in the reference to the earlier work. But we did note there are 30 independent simulations each with its own melt rate.*

*For most of the experiments, we use a randomly generated ensemble of 30 melt distributions applied to the floating nodes (Joughin et al., 2021b), which are used to force 30 independent simulations. Unless otherwise noted, we present the results as the ensemble averages of these simulations.*

*Re time step. Changed to*

*"At each time step (0.01 years),..."*

**L269: I don't quite follow this. Do you mean poorer quality of the velocity used to invert the basal drag coefficient and A?**

*Yes, that's exactly what the sentence says:*

*"All the simulations have some thickening in the upper basin, which is likely due to the poorer quality of the velocity used to initialize the model there (i.e., speeds that are too slow)." The methods section makes it pretty clear this is what we mean by initialize the model.*

**L272-274: I think you need to specify the slight thickening and grounding line advance occur in PIG rather than both glaciers. I saw a few ensembles show more GL retreat in Weertman case (Fig 9f) compared with RCFi case (Fig 9b). Similar things also occur by comparing Fig 9h and Fig 9d. Why is it?**

*Changed from "...near the grounding line..." to "...near the PIG grounding line..."*

*Its not clear why: while we note the differences are small between sliding models and the melt explains much of the variance, there is still some variance due to melt/friction law.*

**L271-274: All of these are talking about PIG so it's better to specify it. I think the velocity contours in Fig8 is distracting to tell the VAF loss near the GL, which is important. When you say 'consistent with Fig7', it's hard to tell the thickening from Fig 7.**

*We added another "PIG" to make it clear that it's the PIG grounding line.*

*We feel the velocity contours are important for showing where the fast flow is, so we have left them in place.*

**L284-285: Then what is causing the lowest VAF loss from Weertman with $h_T$ = 172 at low melt level cases (57 Gt/yr) for PIG compared with other $h_T$?**

*It seems to be related to the fact that when thickening occurs, the bed gets stronger. We changed the text to say.*

*"As noted above, there are limited areas ($h_o < h_T$) where the bed can strengthen if thickening rather than thinning occurs. Such thickening rarely occurs because the region near the grounding line tends to nearly always thin. For some Weertman cases, however, thickening and advance do occur for sufficiently low melt, which should be reinforced by thickening-induced strengthening of the bed near the grounding line. This would explain why the losses decline as $h_T$ increases for the low melt Weertman cases on PIG, since the area subject to this type of strengthening expands. Whether this should remain a feature of our model is a subject for future research."*

**L286: In Fig 8, we can clearly tell the difference between RCFi and Weertman for low melt cases (57 Gt/yr and 75 Gt/yr) at the front of Thwaites region (Fig 8a,b and Fig**

8e,f). Similarly in Fig 7c, the dashed line from hT = 41 m and 86 m gave more mass loss than solid line for low melt case (57 Gt/yr).

*This seems to be a comment rather than request for action.*

L287: It will be good if you can show a map of the basal melt distribution for the Thwaites region. Just pick one of the melt realisations to prove what you said here.

*We don't feel showing a single member of the ensemble will clarify much.*

Does this sentence mean that the distribution of melt did affect the grounding line retreat in Thwaites, which conflicts with your statement that it is not sensitive to the spatial distribution of basal melt.

*The r2 values give the sensitivity. In the case with the least sensitivity, absolute melt explains about 60% of the variance, which represents the majority.*

L290-291: how do you get the 20% and 50%?

*If you look at the figure, the RCF and Weertman results are generally within about 20% of each other. But if you look at the low melt PIG cases as is pointed out, clearly the differences can be greater than 50%.*

L297-301: This comparison between this study and others in the same regions are important. I suggest a figure to compare the basal drag between their regularized Coulomb friction and RCFi in this study for the fast-flowing regions. Again, I don't understand how you decide they produce Coulomb friction for regions where h-h$_f$ < 86 m?

*The 86 is not a precise number, we are just using that contour visual indicator of of the approximate extend where Coulomb behavior can occur.  As noted in the basal friction section, the transition from Coulomb to transition should generally lie below about 67 m.*

L304: Equation (6)?

*Yes, this is a case of Weertman friction explicitly parameterized by effective pressure.*

L310: basal drag of the area near the grounding line is weaker rather than 'area is weaker'.

*The existing text seems fine. You can have a weak area or a strong area.*

L317-318: From Fig 7, the diverge in ice loss for Thwaites is less compared with PIG at low melt values. It's not 'nearly the same' to me with a difference of 10 mm sle.

*We clarified that we meant "PIG with RCFi". Those points are tightly clustered.*

**L334-335: Could you further explain how you translate the values of $H_t$ based on of Barnes and Gudmundsson (2022)?**

*"One way to obtain a rough equivalency is to determine the value of $h_T$ that yields equivalent area-integrated traction subject to reduction via the effective pressure dependency in Equation (4) for a given value of $\alpha^2$. "*

*Translating this into an algorithm note the original text said equation 3 instead of 4, which we fixed.*

1) *Given taub, determine area where Coulomb conditions and weakening will occur (ie., area where $h - h_f < \frac{\tau_b}{\alpha^2 \rho_i g}$).*

2) *Integrate taub over this area.*

3) *Now find $h_T$ such that when $\tau_b$ is integrated over area defined by $h - h_f < h_T$ the result equals the result from step 2.*

*As noted, it's not a perfect equivalent. But at least it provides a rough equivalency such that the total traction subject to weakening is the same.*

**L364: Equation (5) ? à Equation (6)**

*Fixed.*

**L373: you refer to Equation (7) here? so confusing.**

*Should have been Eq 6, fixed.*

**L408: it should be 0.96 or greater.**

*Fixed.*

**L796: basis boundaries à basin boundaries?**

*Figure 7.*

**L409: why the regression value for the ensemble data in Fig 10 (dashed lines) is not consistent with Figure 7 (solid lines)?**

*The dashed lines in Figure 10 should be the same lines as Figure 7 solid, though plotted over a different range. The r^2 values should be different. To make this clear, we added to the caption of Figure 10:*

*"The $r^2_{const.melt}$ values show the fraction of variance that the constant melt regression parameters explain for the depth-parameterized melt function simulations."*

*The point being that if you characterize the model as we did the constant melt simulations, the regression parameters do a pretty good job of predicting the results over a broad range of melt values.*

**L427: I think you refer to Figure 10 rather than Figure 9 here.**

*Fixed.*

**L455: citation please.**

*Added reference to Jourdain et al, 2022 ref.*

**L457: But it is also possible that PIG will have higher basal melt than 67+21 = 88 Gt/yr for the second century, which would exceed your 125 Gt/yr.**

*The math here seems unclear. The point though is that 125 Gt/yr is 2-century average. So if the melt in 2000 was 100Gt/yr and it increased by 25Gt/yr per century. Then it would be 150Gt/yr at the end, but the average would be 125 Gt/yr.*

**L464: You mean with $h_T$ =41 m? If this whole section 4.4 is talking about experiments with $h_T$ = 41, it's better to make it clear at the start of this section.**

*We added (ht=41) wherever there was an ambiguity. It was shorter than adding a sentence and we feel clearer.*

**L470: when did the melt reach 220 Gt/yr in Bett et al. (2023)'s model? The end of 2100 or 2200?**

*Neither, around 50 or 60 years. Based on personal communication with the authors, there are some issues in the preprint with the ice dynamics. But the ocean model should be fine. But for a given VAF loss, the cavity should be similar, and thus, the melting from the ocean model. So, the time is not important here (and likely will change for their final paper).*

**L476: the most aggressive parameterized melt rate function for Thwaites is B&G but is 160_700 when you talk about PIG. It's hard to tell it from Fig S2.**

*For comparison, the most aggressive parameterized melt rate function for Thwaites produces an average melt rate of 151 Gt yr$^{-1}$ ($h_T$=41 m; see B&G in Figure 10c).*

*There is also a slight deviation from the linearity. 160_700 produces a bit less melt but a bit more loss. So we added the parenthetical statement "(Note while 160_700 yields less melt, it produces a slightly large loss for Thwaites.)"*

**L795: Figure 8, for those who is not family with PIG and Thwaites, it's hard to tell the corresponding values of the velocity contours.**

*At the resolution of the plot, they are not intended to be completely discernible. Rather they are to show where fast flow is concentrated. For example, the thinning on Thwaites is much stronger to the east of the fast-flowing trunk, contrary to what one might expect.*

**L797: I guess you refer to Figure 9 here.**

*Yes.*

**L800: Figure 9, is the red line showing the location of the grounding line? What are the scattered points in Fig 9c and 9d?**

*It looks like "lakes" formed for a small number of ensemble members.*

**Figure S2: mr_4 in the legend but mr_2 in the caption?**

*Fixed.*

**References From Reviewer**

Arthern, R. J., and C. R. Williams (2017), The sensitivity of West Antarctica to the submarine melting feedback, *Geophys. Res. Lett.*, 44, 2352–2359, doi:10.1002/2017GL072514.

Gagliardini, O., G. Durand, T. Zwinger, R. C. A. Hindmarsh, and E. Le Meur (2010), Coupling of ice-shelf melting and buttressing is a key process in ice-sheets dynamics, *Geophys. Res. Lett.*, 37, L14501, doi:10.1029/2010GL043334.

Goldberg, D. N., Gourmelen, N., Kimura, S., Millan, R., & Snow, K. (2019). How accurately should we model ice shelf melt rates? *Geophysical Research Letters*, 46, 189–199. https://doi.org/10.1029/2018GL080383

Morlighem, M., Goldberg, D., Dias dos Santos, T., Lee, J., & Sagebaum, M. (2021). Mapping the sensitivity of the Amundsen Sea Embayment to changes in external forcings using automatic differentiation. *Geophysical Research Letters*, 48, e2021GL095440. https://doi.org/10.1029/2021GL095440

Reese, R., Albrecht, T., Mengel, M., Asay-Davis, X., and Winkelmann, R.: Antarctic sub-shelf melt rates via PICO, The Cryosphere, 12, 1969–1985, https://doi.org/10.5194/tc-12-1969-2018, 2018.

---

## Author Comment (AC2)

**Response to Reviewer 1 (Tyler Pelle)**

In this work, Joughin et al. investigate the impact of using both Weertman-style and regularied Coulomb friction law, in conjunction with a linear scaling to the associated basal shear stress field for each law that enhances bed weakening with proximity to the grounding line, on 200-year simulations of Pine Island and Thwaites Glaciers. In addition, this paper also compares the response of these glaciers to an ensemble of randomly generated ice shelf basal melt volumes. The authors find that the choice of friction law has a relatively minor impact on ice volume changes of this sector and recommend use of a regularized Coulomb friction law when only one friction type is used. In addition, parameterized bed weakening led to significant enhancements in the 200-year global sea level contribution of this region, highlighting that such weakening should be included in future ice sheet model simulations either through this explicit manor, or through an effective pressure dependance. Lastly, the sea level response of the system was found to have a strong linear dependance on the total integrated ice shelf melt volume through the 200-year simulations.

*Summary so no response.*

I have to admit that I had a bit of a tough time working my way into this paper because the basal friction overview is quite long and involved, which I think might be a bit much and could be simplified to only include information that is needed to support the results/discussion/conclusions. I also found that visualizing differences in some of the figures was challenging because of the use of different y-axes limits, many of which I think can be made consistent. However, once into the results, I found this work to be an absolute pleasure to read and is full of really wonderful conclusions and insights that would be of wide interest to both the ice sheet modeling and broader scientific communities. It is also clear that this manuscript has been built on a long line of research that the authors have been working on for quite some time, so it is great to see everything come together in such a wonderful way. Below, I provide a number of suggested edits the authors can make to improve the manuscript, most of which aside from restructuring are small-technical corrections that should be easy for the authors to address. Due to the restructuring of the beginning, I suggest major revisions; however, I am very supportive of publication once these comments have been addressed.

*This summarizes comments that we handle in detail below.*

General Comments:
● Basal Friction Overview section: This section is quite long and I have to admit that I got

a bit confused reading through it, which was challenging because it sets the stage for the rest of the paper. There's a lot of analysis and equations presented and I am wondering if it is possible to shorten this section to only what is necessary for interpretation of results/conclusions? After reading through the paper, I think the critical information needed in this section is that *a linear scaling that depends on a height above flotation threshold h_{T} (found in previous work to be ~41-46 m for PIG) is applied to \tau_{b}, which is solved for via both RCF and Weertman-style friction law. In this paper, we will investigate how variation in h_{T} applied to both the BCF and Weertman sliding laws, as well as ice shelf melt, impacts future ice loss of PIG and THW.*

> *To some extent this section is review, but it also provides context for how our implementation relates to other implementations. Since there are a lot of regularized Coulomb friction sliding laws, it is important to examine what the properties are and why they make a difference. As we demonstrate, these laws only provide Coulomb behavior over a small part of the domain (<1%). The real difference they make has more to do with the assumptions made about effective pressure and how that leads to a weakening of the bed as the ice approaches flotation. It's there for the interested reader, but this part can be bypassed by those not interested. We refer back to much of it in the discussion.*

> *We will do another editing pass to see if it can be word-smithed to more digestible.*

Given that, I think both friction laws that are used in this paper, as well as the linear scaling, should be provided, and other information should be limited as needed for clarity. For example, there are a few forms of Weertman sliding laws presented (eqn. 1, 6) and I am not sure which one is used in the simulations.

> *For Weertman we use Equation 1. To make this clear, in the modeling description we added.*

> *"For both the RCFi and Weertman (Equation 1 with $m = 3$), we scale the basal shear stress by $\lambda(h)$, using a range of $h_T$."*

> *The point of introducing Equation 6 is to remind the reader that effective pressure can also be included as part a Weertman type sliding law, causing a similar type of weakening as Equations 3&4.*

Also, the friction law that combines Coulomb and Weertman (eqn. 4) is not used in the rest of the paper, but stimulates quite a long discussion about h_{f} transition points (L81-89) and effective pressure (L91-109; which is not used in either of the friction laws used in this study since it is subsumed into the friction coefficient solution). While I think this information is really fascinating and I think the authors did a great job on the analysis, I unfortunately don't think it is appropriate in its current place in the manuscript and suggest the authors revise this section and shorten it considerably. Perhaps a lot of this can go into an

appendix, with information in the main manuscript saved for only what is most pertinent?

*We disagree because it's important to demonstrate what the friction law we use has in common with other regularized Coulomb friction laws and how they differ. While these equations have been published elsewhere, we feel this section puts them all in the appropriate context.*

*In general, all of the equations are based on sound theory and are affected by (or motivated by) the effective pressure. If we knew the effective pressure everywhere, things would be much better. But since we don't assumptions are made, leading to different behaviors. We feel its worth spending a few words to explore the implications of those assumptions. There are 6 references to Equation 4 scattered throughout the paper.*

● Consistency of language and figure axes in manuscript: I noticed many different forms of "PIG and Thwaites Glaciers", "PIG and Thwaites glaciers", and "Pine Island and Thwaites glaciers". In my opinion, I feel like the "PIG and Thwaites Glacier" would be the most correct, but I am very happy for the authors to choose their favorite variation and use only it throughout the main manuscript, supplement, and in figures and associated captions. I pointed out a few places in the manuscript where I noticed this, but I likely did not catch them all, which is why I raised it as a general issue.

*Good suggestion. We changed the text to remove "PIG and Thwaites glaciers". Now we use either "Pine Island and Thwaites glaciers" or "PIG and Thwaites Glacier" as either is correct. We also removed instances of ...Thwaites... and replaced them with ...Thwaites Glacier.... (there were a couple of instances where Thwaites Glacier was too much to fit in the legend).*

On a similar note, I also noticed many variations of y-axes limits on figure panels that could be made consistent across the entire figure. This could help visualize differences in figures where there are many intersecting lines and many panels.

*Some of these we were able to harmonize as suggested. Others we could not either because it compressed the data in some too much (not much point in comparing between plots if one can read individual plots) or it caused problems with legends.*

**Line Comments**:

● L8: Specify ice shelf basal melt here so readers know you are not referring to melt of grounded ice.

*Done.*

- **L11: Change "above" tp "upstream of"**

  *Done.*

- **L14: remove "work" – i.e. "our simulations suggest"**

  *That wording was grating! Done.*

- **L26: "continued melt forcing" – I think you are referring to ocean induced ice shelf basal melt based on the citations, but I think it would be helpful for readers if you were explicit about this here. Perhaps here you can say " . . . continued ocean forcing in the form of ice shelf basal melting (hereon referred to as melt)."**

  *Done*

- **L33-34: I think it would improve readability if the definition of variables (\tau_{b} and u_{b}) is confined to the "Basal Friction Overview" section. 2**

  *Done*

- **L40: By models, do you mean basal friction parameterizations? If so, please specify because you use "models" twice in this sentence that means two different things.**

  *Agreed. Done.*

- **L50-54: As per line comment L33-34, I think this would be a perfect place to introduce variables \tau_{b} and u_{b}.**

  *Done.*

- **L51: Is this the Weertman sliding law used in the simulations, or is it eqn. 6?**

  *Yes. In response to reviewer 2's comment, we made this clear in the model description.*

- **L70-90: I think there is a lot of interesting information here but it is quite lengthy. If the main point is that we expect Coulomb sliding behavior near the GL and at some transition point, we expect Weertman sliding, maybe this can be said more succinctly with less analysis? This seems very in-depth for a section that is not either the results or discussion.**

  *We refer back to this section multiple times in the discussion where we describe our work relative to other sliding laws. As described above, we feel it provides an important context with which to frame our results. Other than minor word-smithing we have not changed this text.*

  *We feel it is important to emphasize the extent to which Coulomb conditions occur (<1%) given Equations 3&4 are referred to as regularized Coulomb*

*friction. (Granted our version of RCF and RCFi are just over 10%, but they do provide Coulomb behavior over all the fast moving areas).*

● **L81-89: I have to admit that I got a bit lost with some of the analysis here. In particular, I am a little confused about where the value of 45 m came from and why this is only computed for the near GL region (Fig. 1a)**

*To make this more clear, we reworked to*

*"In these examples, Weertman conditions are found everywhere except for the case where Equation (3) is plotted using a height above flotation $(h - h_f)$ of 45 m (Figure 1a)."*

*The value 45 m was selected to show the transition from Coulomb to Weertman. We updated the plot to include 40 m as well, by which point the equation yields nearly full-on Coulomb conditions. We also removed the Equation (4) curves since they only show Weertman in these examples.*

**- can a similar value be computed and added to the plots for trunk (Fig. 1b) and inland tributary (Fig. 1b)?**

*No, because these points are meant to represent inland conditions where Coulomb conditions will never occur (unless the ice thins by several hundred meters).*

*We tried to make this clearer by splitting the paragraph and starting with*

*"The reason why these plots largely reflect Weertman sliding is that the transition from Coulomb to Weertman conditions in Equation (4) occurs at $h - h_f = \frac{\tau_b}{\alpha^2 \rho_i g}$, with Equation (3) producing a less abrupt transition at a similar value. Thus, if we assume ~300 kPa as the maximum expected value for $\tau_b$ with Coulomb friction, then the transition to Weertman sliding takes place at locations where the elevation is less than ~67 m above flotation for $\alpha^2$=0.5."*

**Also, in figure 2, are you plotting h-h_{f} as per the equation at the end of line-83? If so, this should be mentioned in this paragraph and also in the figure caption of figure 2. For figure 2, why are the values of h-h_{f} (1, 41, 86, 176) chosen?**

*First, 176 should be 172, which we have fixed in the new plot. Since it is felt that the text is already too long, rather than explaining this in the text, we added the following to the figure caption.*

*"Note the contour values correspond to the value of $h_T$ used in our simulations."*

● **L93-95: There are recent modeling efforts that model upstream Thwaites/PIG effective pressure with subglacial hydrology models (e.g., Hager et al., 2022; Dow et al. 2023) that show low effective pressure far upstream of the grounding line - I**

would recommend citing them here since they support your claim. It is difficult to know how accurate these model simulations are, but they are likely the best we can do at this point!

*Thanks for pointing us to these references, which we were unaware of. We have cited them as suggested.*

● L129: "Close to the grounding line" is defined as $h-h_{f}<41$, but figure-2 shows that there are numerous regions where the contour of $h-h_{f}=86$ is nearly superimposed onto the line $h-h_{f}=41$; why was the value of 41 m chosen?

*The 41 m value is based on our preferred hT value, which I hope is more clear since we amended the caption.*

*We amended it to "The results show that in the band closest to the grounding line"*

*We are discussing the integrated area between contours, so the fact that the contours are closely spaced in regions doesn't change that.*

● L135: When you say "as the surface elevation approaches flotation", are you referring to in figure 3 when $h-h_{f}$ approaches 0, meaning when $h=h_{f}$? In line 78, you defined h as the ice thickness, so is surface elevation accurate? Perhaps just saying "as ice approaches flotation . . ." would be more clear?

*That was a typo in 78, it should have said elevation not thickness, which we have corrected.*

● L140-145: While I think it is obvious in eqn. 8, I think it should be reiterated that this linear transformation applied to \tau_{b} evolves in time to the changing ice thickness in your simulations.

*Changed* to "…*function produces linear weakening as the surface elevation evolves time similar…"*

● L161: You have two section-2's (the friction overview and methods section).

*Fixed. My bad, but it would be nice if they made the template auto-number the headings.*

● L177: Need a new paragraph space between these paragraphs

*Done. Another thing a properly structured template should have done by including space rather than needing a blank line.*

● L178: Do you mean Equation 7?

*Fixed.*

● **L186: I think it is worth mentioning that your computed fields for the friction coefficient and A do not change in time.**

*"Both A and $\beta^2$ remain constant with time throughout each simulation."*

● **L197: Is the ice front fixed (i.e. no calving is simulated)? Is this a major limitation given that your 2021a work indicated that ice shelf retreat was the primary driver of recent speedup of Pine Island Glacier? If so, I think this should be mentioned somewhere in the text.**

*It's mentioned in the abstract. "Based on recent estimates of melt from other studies, our simulations suggest that melt-driven combined sea-level rise contribution from both glaciers may not exceed 10 cm by 2200, though the uncertainty in model parameters allows for larger increases. We do not include other factors, such as ice shelf breakup that might increase loss, nor factors such as increased accumulation and isostatic uplift that may mitigate loss."*

*And the conclusion "While we can't account for other factors that might increase ice loss, such as full ice shelf breakup <small>(MacAyeal et al., 2003)</small> or partial shelf loss <small>(Joughin et al., 2021a)</small>, our results suggest some bounds on melt-driven losses from PIG and Thwaites Glacier over the next two centuries likely will not exceed 10 cm."*

*We did add:*

*"The domain extent is fixed and the ice-shelf front does not move, but the grounding line evolves freely."*

● **L199: I'm just noticing this now, but I think it would be more correct to say "PIG and Thwaites Glacier" since PIG stands for Pine Island Glacier. See general note about consistency through manuscript.**

*Done.*

● **L208-215: This compiled velocity map is very interesting, it would be great to see a figure of it somewhere in this paper! Perhaps the same can be done for the SMB and combined into one figure?**

*As a figure, it wouldn't be notably different than the plethora of velocity maps already published.*

● **L220: Does SMB vary in time in the simulations?**

*Added ...(Medley et al, 2014), which does not vary with time."*

● **L240-245: Another interesting pattern in figure-5 is that at higher melt rates (panels c/d), corresponding Weertman and RCFi mass loss time-series seem to align more than for lower melt simulations (in the top two panels, the Weertman**

lines are all clumped above the RCFi lines). That is, it seems like the choice of sliding parameterization becomes less important when the system is forced with higher melt rates. Do you agree? This could be an interesting point for the discussion.

*We dug into this a bit deeper prompted by reviewer 2's comments.*

*"As noted above, there are limited areas ($h_0 < h_T$) where the bed can strengthen if thickening rather than thinning occurs. Such thickening rarely occurs because the region near the grounding line tends to nearly always thin. For some Weertman cases, however, thickening and advance do occur for sufficiently low melt, which should be reinforced by thickening-induced strengthening of the bed near the grounding line. This would explain why the losses decline as $h_T$ increases for the low melt Weertman cases on PIG, since the area subject to this type of strengthening expands. Whether this should remain a feature of our model is a subject for future research.*

*We think this addresses the comment here.*

● **L250: Please change "Thwaites" to "Thwaites Glacier" here and also through the manuscript. Also, there are times when you capitalize "Glacier", and when you keep it lowercase (i.e. Thwaites glacier, L261) - I think it is correct for it to be capitalized, so please make that consistent throughout the paper as well.**

*We have changed as described above. Concerning capitalization, we use either Thwaites Glacier or Pine Island Glacier, but Pine Island and Thwaites glaciers when both are referred to by their full names.*

● **L263: Change "... sensitivity for PIG is -0.24 to -0.51 mm Gt^{-1}yr sle for PIG and ..." to "... sensitivity is -0.24 to -0.51 mm Gt^{-1}yr for PIG and ..." (you said "For PIG" twice here)**

*Done.*

● **L270: Is the unit supposed to be "a few hundred m/yr" or "a few hundred m", which seems more in line with the values in figure-8.**

*It should be "m", error due to force of habit for someone who works with velocity a lot.*

● **L277: Change "above the grounding line" to "upstream of the grounding line" here and throughout the manuscript.**

*Done.*

● **L281: The notation "Figure 9e&f" seems a little messy, perhaps could use "Figure e/f"? I think this is personal preference and maybe a little pedantic, but I have not seen the "&" symbol used in this manor in a manuscript before.**

*"&" means "and" and is commonly used by journals in this way. We can't find an EGU style guide to resolve the issue, so will leave it to the copy editors.*

● **L287-289: It would be nice to see what some of these melt distributions look like since they seem like a primary control on the spatial distribution of retreat of Thwaites Glacier.**

*Rather than including examples here, we added: "(see example profiles in Joughin et al, 2021b)."*

● **L317-318: I would even say that simulations with greater $h_{T}$ values lose less mass than those with higher $h_{T}$ values for these PIG simulations, which is pretty fascinating! I really enjoy your call to earlier work here that investigated this paradox.**

*We are taking this as a comment and not a change request.*

● **L346: I'm not sure if it is just the way my computer rendered the PDF, but it seems like there is an unnecessary space in the word "ri ght", but I'm not sure if this is a typo.**

*Weird. It looks fine in the Word doc.*

● **L350-357: I really enjoy this discussion and am excited with the prospect of future work that compares such implementations of bed weakening near the grounding line.**

*Thanks.*

● **L359-374: I understand the reduction of section "Basal Friction Overview" might cause issues with this and the preceding discussion paragraph, but I think if you put a lot of the**

**"Basal Friction Overview" information in an appendix, you can still keep these paragraphs (with references to the equations in the appendix), which I think are very important findings and thoughts from your work.**

*For the reasons discussed above, we think the text is best left where it is.*

● **L377: I've noticed a few variations of "PIG and Thwaites Glacier" (L377), "PIG and Thwaites glaciers" (L261), and "Pine Island and Thwaites glaciers" (L770). Please pick one and keep consistent throughout the manuscript, supplement, and all figures and figure captions.**

*We have made the capitalization consistent as suggested. Stylistically, we prefer to alternate in places between "PIG and Thwaites Glacier" and "Pine Island and Thwaites glaciers".*

● **L406-417: See note for figure 10 below - In short, I would recommend a different way of showing these results. Given that this paragraph focuses mainly on the associated r^{2} values, I'm wondering if figure 10 could be consolidated into a table?**

*We feel it's better to display the results graphically.*

● **L427-429:I got confused by the phrasing "results lie along a line" and "results that fall well off a line"; it took me a second to realize you are referring to a linear regression line. Maybe rephrase to clarify.**

*Reworded. "For example, plotting results from multiple studies as shown in Figure 10 would help differentiate the cases where different models produce results consistent with the level of melt forcing (e.g., the results lie along a linear regression line with high $r^2$ near 1) from those where the differences are due to some other aspect of the model (e.g., results are not explained well by a linear regression to melt)."*

● **L430-432: Seroussi et al. (2023) found that treatment of ice dynamics was the main driver of uncertainty in the ISMIP6 ensemble through 2100; however, this was across the entire AIS. In line-431, maybe specify that you are only referring to the ASE (i.e. ". . . suggests that differences between models in this sector may largely . . .").**

**The existing sentence did refer to the ASE, but we made more clear with**

*"For example, the fact that melt is the main predictor of loss in the Amundsen Sea Embayment for the suite of ISMIP6 models (Seroussi et al., 2020), suggests that much of the difference between models in this region may be due to how they treat melt, as opposed to differences in their treatment of ice dynamics.*

● **L440: This is true for the ASE, but would these conclusions hold for other sectors of Antarctica? If you are not sure, it might be worth specifying here that you are referring to coupled ice-ocean models of the ASE.**

*We don't want to clutter the discussion with more detail on this matter. But since you are interested, see Figure 5 in Joughin et al., 2021b.*

● **L502: Consider rewording: " . . . our results suggest that melt-driven losses from PIG and Thwaites Glacier over the next two centuries likely will not exceed 10 cm."**

*Softened it a bit. But we are clear this is not all source of loss, only melt-driven. It now reads.*

*"While we can't account for other factors that might increase ice loss, such as full ice shelf breakup (MacAyeal et al., 2003) or partial shelf loss (Joughin et al., 2021a), our results suggest melt-driven losses from PIG and Thwaites Glacier over the next two centuries may not exceed 10 cm. Two centuries out, however, both glaciers will have lost a substantial amount of ice and will be primed for much more rapid loss if melt rates don't subside. "*

● **L511-512: Please remove hyperlinks – also, I think "comit" should be changed to "commit" in this section.**

*The http addresses are needed. The template is making them hyperlinks, which should get taken care of when the final ms is typeset (some journals seem to keep them as hyperlinks).*

● **L797: Magenta box in panel-a denotes domain for figure 9 (not figure 10)**

*Fixed.*

**Figure Comments**:

● **It can be quite challenging to compare VAF losses between panels in figures 4-7 since the y-axis limits are all different. Where appropriate, can you please use the same y-axis limits? I think this would be very helpful for all panels in figures 4-6, as well as panels B/C of figure 7. Otherwise, all of the curves look fairly similar and it can be difficult to visualize differences between them, which I think is the ultimate goal of these figures.**

*We made some changes, but not all of them. The figures have several lines, which become very crowded when using the max y-range for all plots.*

● **Figure 2: Perhaps it was said in the main text, but it would be good (in the figure caption) to reiterate the friction law that was used to compute the basal shear stress that is shown in panels-A/B.**

*It shouldn't really matter since the RCFi and Weertman should yield the same basal shear stress within minor numerical differences. But we did change to "basal shear stress ($\tau_b$ from RCFi inversion)".*

● **Figure 3: The text and lines in this figure are a little blurry and/or choppy (as with figure 1 as well). Will you output these as PDF's in the final submission?**

*The original figures look great, but we did try to increase some fonts and thicken some lines. I think we provide original figures instead of a word doc for the final.*

- **Figure 8: It doesn't seem like two color bars are needed, but rather you could use one that diverges at 0, with blue (negative) trailing off to the left, and red (positive) trailing off to the right. Also, would it be possible to scale the intensity of the blue so that it uses the same color scale as red (i.e. reaches maximum intensity at -500 m). This should result in very pale shading of blue where you have height above flotation gains, which would seem more appropriate since now, it appears like the gains in the bottom and upstream parts of the domain are stronger than the losses Thwaites Glacier experiences, which is not true.**

> *The thickening is an order of magnitude less than the thinning, so we did it this way so as not to bury any thickening in the color table. We assume the reader can take the different scales into account (having two color bars sort of forces them to notice the scale difference).*

- **Figure 10: I believe I understand that the main point of this figure is to show that the linear relationship between total integrated melt volume and VAF change holds for various other melt parameterizations, but I think the number of different-shaped symbols is very distracting and is rather uninterpretable for the reader. Many of the symbols are clumped over each other at lower x-values and also, the meaning of the different symbols is not explained in the main text (i.e., I know the different symbols are different melt function outputs, but I have no idea what "mr_1", "mr_4", "80_700", etc. are specifically), and I don't think readers should have to read the supplement to interpret a main-text figure.**

> *Issue 1), too many symbols. There are a lot, but we feel it's important to show the scatter of the full range of models (using the same scale as requested above would only make the issue worse). For the most part, they are distinguishable, and when they are crammed together it indicates nearly the same result. We wanted to clarify that the linearity with melt is not some fluke of how we normalized melt in the ensemble simulations.*

> *Issue 2), melt functions in the supplement. We think the casual reader can interpret the figure shows a collection of depth-parameterized melt rates that produce a range of melt, which yield losses that respond linearly. Those really interested can see the supplement. And it's not without precedent, as we had to delve into supplements to get some of the parameters used to create this supplement figure.*

Ultimately, I don't think the symbols matter given that the analysis is not dependent on individual melt functions, so I wonder if the authors can remove the 10 symbol-types and use just one symbol and the different colors to represent the different melt function outputs?

*We argue one can ignore the symbol types and do this anyway. But if someone is really curious about a particular melt parameterization, in most cases they can find it using the symbols (or at least see it falls in a cluster)*

*.*

I per line comment L406-417, I think this figure could also be consolidated into a table given that the main piece of information that is used from it are the r^{2} values.

*Seeing is believing and we feel the regression lines plotted over the points makes a stronger statement.*

● Figure S1: No change is needed in the text, but I am just curious what metric you use to prescribe your mesh resolution?

*We plot the Firedrake function*

[Figure]

*Which should give the long side of each triangle.*

● Figure S2: In the figure caption, please include the full in-text citation for Barnes and Gudmundsson (2022).

*Fixed.*

---

## Author Response (AR1)

The detailed, point-by-point responses are all included in the response to reviewers.

I have prepared the data for archiving at the University of Washington ResearchWorks archive. It will take a few days to get the DOI, which can be inserted in the final version.

My reference software updates the references each time it regenerates the biography, cluttering the tracked changes. I have gone through and manually accepted all of these changes. I could have inadvertently accepted an actual change in the process (unlikely, and if so, it was likely a newly inserted ref).

---

## Author Response (AR2)

Dear Dr McCormack,

Thanks for your time and effort in turning this around quickly.

- Both reviewers had comments around the basal friction law overview in Section 2. I appreciate your response to contrasting suggestions for more detail and shortening of this section! However, to address any confusion, I suggest that a brief overview is added at the start of this section, with a few sentences describing why this section is included in the manuscript. This could include parts of the description in the response to reviewer 2, e.g. :

"A major focus of this paper is to try to systematically separate the effective pressure response (near grounding line weakening) from the friction type (Coulomb versus Weertman)." To this end, in this section, we review/provide context on…

**Done.**

- Please provide a brief overview text in the supplementary material to accompany the figures. For figure S1, could you please comment on how the mesh resolution is determined / refined? Why are there linear features of coarser resolution in the interior of the catchment?

**Done.**

- I note that you have prepared the data for archiving - thanks very much. Do you have a DOI of the data and model inputs? In accordance with TC open access requirements, I would like to check that before approving the final version.

**Active Dryad link provided in revision.**

I likely will be starting a course of chemo and radiation the week after next (prognosis so far is quite good).  If I get the green light to upload before then, I will do my best to get it done ASAP.  Once treatment has started, I maybe a bit slower to response. Thanks in advance for your patience.

Ian